# How Feature Learning Can Improve Neural Scaling Laws

**Blake Bordelon**[*], **Alexander Atanasov**[*] , **Cengiz Pehlevan**

## Abstract

We develop a solvable model of neural scaling laws beyond the kernel limit. Theoretical analysis of this model shows how performance scales with model size, training time, and the total amount of available data. We identify three scaling regimes corresponding to varying task difficulties: hard, easy, and super easy tasks. For easy and super-easy target functions, which lie in the reproducing kernel Hilbert space (RKHS) defined by the initial infinite-width Neural Tangent Kernel (NTK), the scaling exponents remain unchanged between feature learning and kernel regime models. For hard tasks, defined as those outside the RKHS of the initial NTK, we demonstrate both analytically and empirically that feature learning can improve scaling with training time and compute, nearly doubling the exponent for hard tasks. This leads to a different compute optimal strategy to scale parameters and training time in the feature learning regime. We support our finding that feature learning improves the scaling law for hard tasks but not for easy and super-easy tasks with experiments of nonlinear MLPs fitting functions with power-law Fourier spectra on the circle and CNNs learning vision tasks.

## 1 Introduction

Deep learning models tend to improve in performance with model size, training time and total available data. The dependence of performance on the available statistical and computational resources are often regular and well-captured by a power-law (Hestness et al., 2017; Kaplan et al., 2020). For example, the Chinchilla scaling law (Hoffmann et al., 2022) for the loss $\mathcal{L}(t, N)$ of a $N$-parameter model trained online for $t$ steps (or $t$ tokens) follows

$$\mathcal{L}(t, N) = c_t t^{-r_t} + c_N N^{-r_N} + \mathcal{L}_\infty, \tag{1}$$

where the constants $c_t, c_N$ and exponents $r_t, r_N$ are dataset and architecture dependent and $\mathcal{L}_\infty$ represents the lowest achievable loss for this architecture and dataset. These scaling laws enable intelligent strategies to achieve performance under limited compute budgets (Hoffmann et al., 2022) or limited data budgets (Muennighoff et al., 2023). A better understanding of what properties of neural network architectures, parameterizations and data distributions give rise to these neural scaling laws could be useful to select better initialization schemes, parameterizations, and optimizers (Yang et al., 2021; Achiam et al., 2023; Everett et al., 2024) and develop better curricula and sampling strategies (Sorscher et al., 2022).

Despite significant empirical research, a predictive theory of scaling laws for deep neural network models is currently lacking. Several works have recovered data-dependent scaling laws from the analysis of linear models (Spigler et al., 2020; Bordelon et al., 2020; Bahri et al., 2021; Maloney et al., 2022; Simon et al., 2021; Bordelon et al., 2024a; Zavatone-Veth & Pehlevan, 2023; Paquette et al., 2024; Lin et al., 2024). However these models are fundamentally limited to describing the kernel or *lazy learning* regime of neural networks (Chizat et al., 2019). Several works have found that this fails to capture the scaling laws of deep networks in the feature learning regime (Fort et al., 2020; Vyas et al., 2022; 2023a; Bordelon et al., 2024a). A theory of scaling laws that can capture consistent feature learning even in an infinite parameter $N \to \infty$ limit is especially pressing given the success of mean field and $\mu$-parameterizations which generate constant scale feature updates across model widths and depths (Mei et al., 2019; Geiger et al., 2020; Yang & Hu, 2021; Bordelon & Pehlevan, 2022; Yang et al., 2022; Bordelon et al., 2023; 2024b). The training dynamics of the

---

[*]Equal Contribution

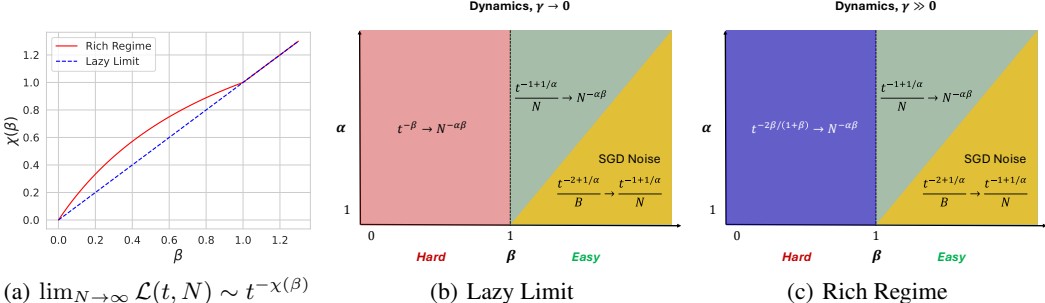

(a) $\lim_{N\to\infty} \mathcal{L}(t, N) \sim t^{-\chi(\beta)}$      (b) Lazy Limit      (c) Rich Regime

Figure 1: Our model changes its scaling law exponents for hard tasks, where the source is sufficiently small $\beta < 1$. (a) The exponent $\chi(\beta)$ which appears in the loss scaling $\mathcal{L}(t) \sim t^{-\chi(\beta)}$ of our model. (b)-(c) Phase plots in the $\alpha, \beta$ plane of the observed scalings that give rise to the compute-optimal trade-off. Arrows ($\to$) represent a transition from one scaling behavior to another as $t \to \infty$, where the balancing of these terms at fixed compute $C = Nt$ gives the compute optimal scaling law. In the lazy limit $\gamma \to 0$, we recover the phase plot for $\alpha > 1$ of Paquette et al. (2024). At nonzero $\gamma$, however, we see that the set of "hard tasks", as given by $\beta < 1$ exhibits an improved scaling exponent. The compute optimal curves for the easy tasks with $\beta > 1$ are unchanged.

infinite width/depth limits in such models can significantly differ from the lazy training regime. Infinite limits which preserve feature learning are better descriptors of practical networks Vyas et al. (2023a). Motivated by this, we ask the following:

**Question:** *Under what conditions can feature learning improve the scaling law exponents of neural networks compared to lazy training regime?*

### 1.1 OUR CONTRIBUTIONS

In this work, we develop a theoretical model of neural scaling laws that allows for improved scaling exponents compared to lazy training under certain settings. Our contributions are

1. We propose a simple two-layer linear neural network model trained with a form of projected gradient descent. We show that this model reproduces power law scalings in training time, model size and training set size. The predicted scaling law exponents are summarized in terms of two parameters related to the data and architecture $(\alpha, \beta)$.

2. We identify a condition on the difficulty of the learning task, measured by the source exponent $\beta$, under which feature learning can improve the scaling of the loss with time and with compute. For easy tasks, which we define as tasks with $\beta > 1$ where the RKHS norm of the target is finite, there is no improvement in the power-law exponent while for hard tasks ($\beta < 1$) that are outside the RKHS of the initial limiting kernel, there can be an improvement. For super-easy tasks $\beta > 2 - \frac{1}{\alpha}$, which have very low RKHS norm, variance from stochastic gradient descent (SGD) can alter the scaling law at large time. Figure 1 summarizes these results.

3. We provide an approximate prediction of the compute optimal scaling laws for hard, easy tasks and super-easy tasks. Each of these regimes has a different exponent for the compute optimal neural scaling law. Table 1 summarizes these results.

4. We test our predicted feature learning scalings by training deep nonlinear neural networks fitting nonlinear functions. In many cases, our predictions from the initial kernel spectra accurately capture the test loss of the network in the feature learning regime.

Overall our results suggest that feature learning may improve scaling law exponents by changing the optimization trajectory for tasks that are hard for the initial kernel.

### 1.2 RELATED WORKS

Our work builds on the recent results of Bordelon et al. (2024a) and Paquette et al. (2024) which analyzed the SGD dynamics of a structured random feature model. Statics of this model have been analyzed by many prior works (Atanasov et al., 2023; Simon et al., 2023; Zavatone-Veth & Pehlevan,

2023; Bahri et al., 2021; Maloney et al., 2022). These kinds of models can accurately describe networks in the lazy learning regime. However, the empirical study of Vyas et al. (2022) and some experiments in Bordelon et al. (2024a) indicate that the predicted compute optimal exponents were smaller than those measured in networks that learn features on real data. These latter works observed that networks train faster in the rich regime compared to lazy training. We directly address this gap in performance between lazy and feature learning neural networks by allowing the kernel features to adapt to the data. We revisit the computer vision settings of Bordelon et al. (2024a) and show that our new exponents more accurately capture the scaling law in the feature learning regime.

Other work has investigated when neural networks outperform kernels (Ghorbani et al., 2020). Ba et al. (2022) and Abbe et al. (2023) have shown how feature learning neural networks can learn low rank spikes in the hidden layer weights/kernels to help with sparse tasks while lazy networks cannot. Target functions with staircase properties, where learning simpler components aid learning of more complex components also exhibit significant improvements (with respect to a large input dimension) due to feature learning (Abbe et al., 2021; Dandi et al., 2023; Bardone & Goldt, 2024). Here, we consider a different setting. We ask whether feature learning can lead to improvements in power law exponents for the neural scaling law. The work of Paccolat et al. (2021) asks a similar question in the case of a simple stripe model. Here we investigate whether the power law scaling exponent can be improved with feature learning in a model that only depends on properties of the initial kernel and the target function spectra. Recent works have examined the dynamics of linear networks, contrasting the dynamics in lazy and feature learning regime, including analysis of infinite width linear networks Chizat et al. (2024), and linear networks with varying and unbalanced initialization and learning rates Kunin et al. (2024); Tu et al. (2024). Our model can be interpreted as a two-layer linear network which captures finite width effects (with task-dependent scaling laws) from random initialization. Like these related works, our model also has unbalanced learning rates between hidden and readout weights set by a parameter $\gamma$ that recovers a lazy limit as $\gamma \to 0$.

## 2 SOLVABLE MODEL OF SCALING LAWS WITH FEATURE LEARNING

We start by motivating and defining our model. Our goal is to build a simple model that exhibits feature learning in the infinite-width limit but also captures finite network size, finite batch SGD effects, and sample size effects that can significantly alter scaling behavior. In this work, our operational definition of feature learning is evolution of the neural tangent kernel (NTK) of the model.[1]

Following the notation of Bordelon et al. (2024a), we introduce our model from the perspective of kernel regression. We first assume a randomly initialized neural network in an infinite-width limit where NTK concentrates. We then diagonalize the initial infinite-width NTK. The resulting eigenfunctions $\boldsymbol{\psi}_\infty(\boldsymbol{x}) \in \mathbb{R}^M$ have an inner product that define the infinite-width NTK $K_\infty(\boldsymbol{x}, \boldsymbol{x}') = \boldsymbol{\psi}_\infty(\boldsymbol{x}) \cdot \boldsymbol{\psi}_\infty(\boldsymbol{x}')$. These eigenfunctions are orthogonal under the probability distribution of the data $p(\boldsymbol{x})$ with

$$\left\langle \boldsymbol{\psi}_\infty(\boldsymbol{x}) \boldsymbol{\psi}_\infty(\boldsymbol{x})^\top \right\rangle_{\boldsymbol{x} \sim p(\boldsymbol{x})} = \boldsymbol{\Lambda} = \mathrm{diag}(\lambda_1, ..., \lambda_M). \tag{2}$$

We will often consider the case where $M \to \infty$ first so that these functions $\boldsymbol{\psi}_\infty(\boldsymbol{x})$ form a complete basis for the space of square integrable functions. We next consider a finite sized model with $N$ parameters. We assume this model's initial parameters are sampled from the same distribution as the infinite model and that the $N \to \infty$ limit recovers the same kernel $K_\infty(\boldsymbol{x}, \boldsymbol{x}')$. The finite $N$-parameter model, at initialization $t = 0$, has $N$ eigenfeatures $\tilde{\boldsymbol{\psi}}(\boldsymbol{x}, 0) \in \mathbb{R}^N$. Unlike the lazy regime, in the feature learning regime, these features will evolve during training.

The finite network's learned function $f$ is expressed in terms of the lower dimensional features, while the target function $y(\boldsymbol{x})$ can be decomposed in terms of the limiting (and static) features $\boldsymbol{\psi}_\infty(\boldsymbol{x})$ with coefficients $\boldsymbol{w}^\star$. The instantaneous finite width features $\tilde{\boldsymbol{\psi}}(\boldsymbol{x}, t)$ can also be expanded as a linear combination of the basis functions $\boldsymbol{\psi}_\infty(\boldsymbol{x})$ with coefficient matrix $\boldsymbol{A}(t) \in \mathbb{R}^{N \times M}$. We can therefore view our model as the following student-teacher setup

$$f(\boldsymbol{x}, t) = \frac{1}{N} \boldsymbol{w}(t) \cdot \tilde{\boldsymbol{\psi}}(\boldsymbol{x}, t), \quad \tilde{\boldsymbol{\psi}}(\boldsymbol{x}, t) = \boldsymbol{A}(t) \boldsymbol{\psi}_\infty(\boldsymbol{x})$$
$$y(\boldsymbol{x}) = \boldsymbol{w}^\star \cdot \boldsymbol{\psi}_\infty(\boldsymbol{x}). \tag{3}$$

---

[1]Other definitions are possible, but NTK evolution is at least a *necessary* condition for feature learning.

If the matrix $\boldsymbol{A}(t)$ is random and static then gradient descent on this random feature model recovers the lazy network analyzed by Bordelon et al. (2024a); Paquette et al. (2024). In this work, we extend the analysis to cases where the matrix $\boldsymbol{A}(t)$ is also updated, to allow for the evolution of the kernel. We consider online training in the main text but discuss and analyze the case where samples are reused in Appendix D.

We allow $\boldsymbol{w}(t)$ to evolve with stochastic gradient descent (SGD) and $\boldsymbol{A}(t)$ evolve by *projected SGD* on a mean square error with batch size $B$. Letting $\boldsymbol{\Psi}_\infty(t) \in \mathbb{R}^{B \times M}$ represent a randomly sampled batch of $B$ points evaluated on the limiting features $\{\boldsymbol{\psi}_\infty(\boldsymbol{x}_\mu)\}_{\mu=1}^B$ and $\eta$ to be the learning rate, our updates take the form

$$\boldsymbol{w}(t+1) - \boldsymbol{w}(t) = \eta \boldsymbol{A}(t) \left( \frac{1}{B} \boldsymbol{\Psi}_\infty(t)^\top \boldsymbol{\Psi}_\infty(t) \right) \boldsymbol{v}^0(t) \,, \; \boldsymbol{v}^0(t) \equiv \boldsymbol{w}_\star - \frac{1}{N} \boldsymbol{A}(t)^\top \boldsymbol{w}(t)$$

$$\boldsymbol{A}(t+1) - \boldsymbol{A}(t) = \eta \gamma \, \boldsymbol{w}(t) \boldsymbol{v}^0(t)^\top \left( \frac{1}{B} \boldsymbol{\Psi}_\infty(t)^\top \boldsymbol{\Psi}_\infty(t) \right) \left( \frac{1}{N} \boldsymbol{A}(0)^\top \boldsymbol{A}(0) \right). \tag{4}$$

The fixed random projection $\left( \frac{1}{N} \boldsymbol{A}(0)^\top \boldsymbol{A}(0) \right)$ present in $\boldsymbol{A}(t)$'s dynamics ensure that the features cannot have complete access to the infinite width features $\boldsymbol{\psi}_\infty$ but only access to the initial $N$-dimensional features $\boldsymbol{A}(0)\boldsymbol{\psi}_\infty$. If this term were not present then there would be no finite parameter bottlenecks in the model and even a model with $N = 1$ could fully fit the target function, leading to trivial parameter scaling laws[2]. In this sense, the vector space spanned by the features $\tilde{\boldsymbol{\psi}}$ *does not change* over the course of training, but the finite-width kernel Hilbert space *does change* its kernel: $\tilde{\boldsymbol{\psi}}(\boldsymbol{x}, t) \cdot \tilde{\boldsymbol{\psi}}(\boldsymbol{x}', t)$. Feature learning in this space amounts to reweighing the norms of the existing features. We have chosen $\boldsymbol{A}(t)$ to have dynamics similar to the first layer weight matrix of a linear neural network. As we will see, this is enough to lead to an improved scaling exponent.

The hyperparameter $\gamma$ sets the speed of $\boldsymbol{A}$'s dynamics and thus controls the rate of feature evolution. The $\gamma \to 0$ limit represents the *lazy learning* limit Chizat et al. (2019) where features are static and coincides with a random feature model dynamics of Bordelon et al. (2024a); Paquette et al. (2024). The test error after $t$ steps on a $N$ parameter model with batch size $B$ is

$$\mathcal{L}(t, N, B, \gamma) \equiv \left\langle \left[ \boldsymbol{\psi}_\infty(\boldsymbol{x}) \cdot \boldsymbol{w}^* - \tilde{\boldsymbol{\psi}}(\boldsymbol{x}, t) \cdot \boldsymbol{w}(t) \right]^2 \right\rangle_{\boldsymbol{x} \sim p(\boldsymbol{x})} = \boldsymbol{v}^0(t)^\top \boldsymbol{\Lambda} \boldsymbol{v}^0(t). \tag{5}$$

In the next sections we will work out a theoretical description of this model as a function of the spectrum $\boldsymbol{\Lambda}$ and the target coefficients $\boldsymbol{w}^\star$. We will then specialize to power-law spectra and target weights and study the resulting scaling laws.

## 3 DYNAMICAL MEAN FIELD THEORY OF THE MODEL

We can consider the dynamics for random $\boldsymbol{A}(0)$ and random draws of data during SGD in the limit of $M \to \infty$ and $N, B \gg 1$[3]. This dimension-free theory is especially appropriate for realistic trace class kernels where $\langle K_\infty(\boldsymbol{x}, \boldsymbol{x}') \rangle_{\boldsymbol{x}} = \sum_k \lambda_k < \infty$ (equivalent to $\alpha > 1$), which is our focus. Define $w_k^\star, v_k(t)$ to be respectively the components of $\boldsymbol{w}^\star, \boldsymbol{v}^0(t)$ in the $k$th eigenspace of $\boldsymbol{\Lambda}$. The error variables $v_k^0(t)$ are given by a stochastic process, and yield deterministic prediction for the loss $\mathcal{L}(t, N, B, \gamma)$, analogous to the results of Bordelon et al. (2024a).

Since the resulting dynamics for $v_k^0(t)$ at $\gamma > 0$ are nonlinear and cannot be expressed in terms of a matrix resolvent, we utilize dynamical mean field theory (DMFT), a flexible approach for handling nonlinear dynamical systems driven by random matrices (Sompolinsky & Zippelius, 1981; Helias & Dahmen, 2020; Mannelli et al., 2019; Mignacco et al., 2020; Gerbelot et al., 2022; Bordelon et al., 2024a). Most importantly, the theory gives closed analytical predictions for $\mathcal{L}(t, N, B, \gamma)$. We defer the derivation and full DMFT equations to the Appendix C. The full set of closed DMFT equations are given in Equation equation 26 for online SGD and Equation

---

[2]We could also solve this problem by training a model of the form $f = \boldsymbol{w}(t)^\top \boldsymbol{B}(t) \boldsymbol{A} \boldsymbol{\psi}$ where $\boldsymbol{w}(t)$ and $\boldsymbol{B}(t)$ are dynamical with initial condition $\boldsymbol{B}(0) = \boldsymbol{I}$ and $\boldsymbol{A}$ frozen and the matrix $\boldsymbol{B}(t)$ following gradient descent. We show that these two models are actually exhibit equivalent dynamics in Appendix B.

[3]There are finite size fluctuations around the mean-field at small $N, B$ which are visible in errorbars in Figure 2 (c)-(d), which could also be extracted from the theory. Alternatively, we can operate in a proportional limit with $N/M, B/M$ approaching constants, which is exact with no finite size fluctuations.

equation 41 for offline training with data repetition. Informally, this DMFT computes a closed set of equations for the correlation and response functions for a collection of time-varying vectors $\mathcal{V} = \{\boldsymbol{v}^0(t), \boldsymbol{v}^1(t), \boldsymbol{v}^2(t), \boldsymbol{v}^3(t), \boldsymbol{v}^4(t)\}_{t \in \{0,1,\dots\}}$ including $C_0(t,s) = \boldsymbol{v}^0(t)^\top \boldsymbol{\Lambda} \boldsymbol{v}^0(s)$ which directly gives the test loss $\mathcal{L}(t) = C_0(t,t)$. This theory is derived generally for any spectrum $\lambda_k$ and any target $w_k^\star$. In the coming sections we will examine approximate scaling behavior of the loss when the spectrum follows a power law. In the figures, we will plot the predictions from the full DMFT equations as dashed black lines.

## 4 POWER LAW SCALINGS FROM POWER LAW FEATURES

We consider initial kernels that satisfy source and capacity conditions as in (Caponnetto & Vito, 2005; Pillaud-Vivien et al., 2018; Cui et al., 2021; 2023). These conditions measure the rate of decay of the spectrum of the *initial infinite width* kernel $K_\infty(x, x')$ and target function $y(x)$ in that basis. Concretely, we consider settings with the following power law scalings:

$$\lambda_k \sim k^{-\alpha}, \qquad \sum_{\ell > k} \lambda_\ell (w_\ell^*)^2 \sim k^{-\alpha\beta}. \tag{6}$$

The exponent $\alpha$ is called the **capacity** and measures the rate of decay of the initial kernel eigenvalues. We will assume this exponent is greater than unity $\alpha > 1$ since the limiting $N \to \infty$ kernel should be trace class. The exponent $\beta$ is called the **source** and quantifies the difficulty of the task under kernel regression with $K_\infty$.[4] The RKHS norm $|\cdot|_{\mathcal{H}}^2$ of the target function is given by:

$$|y|_{\mathcal{H}}^2 = \sum_k (w_k^\star)^2 = \sum_k k^{-\alpha(\beta-1)-1} \approx \begin{cases} \frac{1}{\alpha(\beta-1)} & \beta > 1 \\ \infty & \beta < 1. \end{cases} \tag{7}$$

While the case of finite RKHS norm ($\beta > 1$) is often assumed in analyses of kernel methods that rely on norm-based bounds, such as (Bartlett & Mendelson, 2002; Bach, 2024), the $\beta < 1$ case is actually more representative of real datasets. This was pointed out in (Wei et al., 2022). This can be seen by spectral diagonalizations performed on real datasets in (Bahri et al., 2021; Bordelon et al., 2024a) as well as in experiments in Section 5.2. We stress this point since the behavior of feature learning with $\beta > 1$ and $\beta < 1$ will be strikingly different in our model.

**General Scaling Law in the Lazy Limit** For the purposes of deriving compute optimal scaling laws, the works of Bordelon et al. (2024a) and Paquette et al. (2024) derived precise asymptotics for the loss curves under SGD. For the purposes of deriving compute optimal scaling laws, these asymptotics can be approximated as the following sum of power laws at large $t, N$

$$\lim_{\gamma \to 0} \mathcal{L}(t, N, B, \gamma) \approx \underbrace{t^{-\beta}}_{\text{Limiting Gradient Flow}} + \underbrace{N^{-\alpha \min\{2, \beta\}}}_{\text{Model Bottleneck}} + \underbrace{\frac{1}{N} t^{-(1 - \frac{1}{\alpha})}}_{\text{Finite } N \text{ Transient}} + \underbrace{\frac{\eta}{B} t^{-(2 - \frac{1}{\alpha})}}_{\text{SGD Transient}}. \tag{8}$$

where we neglect prefactor constants that are independent of $t, N, B$ (though these can be extracted from the full theory). The first terms represent *bottleneck/resolution-limited scalings* which represent the loss obtained by taking all but one of the scaling quantities to infinity (Bahri et al., 2021). The first term gives the loss dynamics of population gradient flow ($N, B \to \infty$) while the second (model bottleneck) term describes $t \to \infty$ limit of the loss which depends on $N$. The third and fourth terms are mixed *transients* that arise from the perturbative finite model and batch size effects. While Bordelon et al. (2024a) focused on *hard tasks* where $\beta < 1$ where the first two terms dominate when considering compute optimal scaling laws, Paquette et al. (2024) also discussed two other phases of the easy task regime $1 < \beta < 2 - 1/\alpha$ where the first and third term dominate and the super easy regime $\beta > 2 - 1/\alpha$ where the final two terms dominate the compute optimal scaling.

**General Scaling Law in the Feature Learning Regime** For $\gamma > 0$, approximations to our precise DMFT equations under power law spectra give the following sum of power laws

$$\mathcal{L}(t, N, B, \gamma) \approx \underbrace{t^{-\beta \max\{1, \frac{2}{1+\beta}\}}}_{\text{Limiting Gradient Flow}} + \underbrace{N^{-\alpha \min\{2, \beta\}}}_{\text{Model Bottleneck}} + \underbrace{\frac{1}{N} t^{-(1 - \frac{1}{\alpha}) \max\{1, \frac{2}{1+\beta}\}}}_{\text{Finite } N \text{ Transient}} + \underbrace{\frac{\eta}{B} t^{-(2 - \frac{1}{\alpha}) \max\{1, \frac{2}{1+\beta}\}}}_{\text{SGD Transient}}.$$

$$\tag{9}$$

---

[4]The source exponent $r$ used in (Pillaud-Vivien et al., 2018) and other works is given by $2r = \beta$.

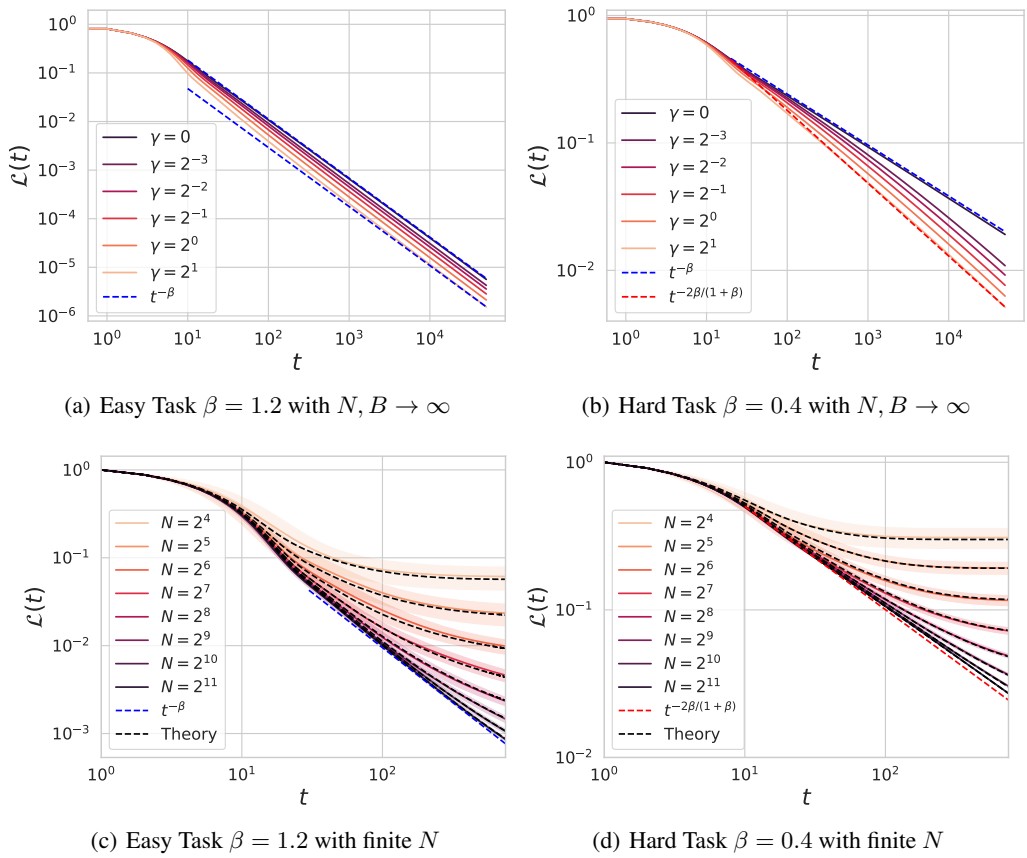

Figure 2: The learning dynamics of our model under power law features exhibits power law scaling with an exponent that depends on task difficulty. Dashed black lines represent solutions to the dynamical mean field theory (DMFT) while colored lines and shaded regions represent means and errorbars over 32 random experiments. (a) For easy tasks with source exponent $\beta > 1$, the loss is improved with feature learning but the exponent of the power law is unchanged. We plot the approximation $\mathcal{L} \sim t^{-\beta}$ in blue. (b) For hard tasks where $\beta < 1$, the power law scaling exponent improves. An approximation of our learning curves predicts a new exponent $\mathcal{L} \sim t^{-\frac{2\beta}{1+\beta}}$ which matches the exact $N, B \to \infty$ equations. (c)-(d) The mean field theory accurately captures the finite $N$ effects in both the easy and hard task regimes. As $N \to \infty$ the curve approaches $t^{-\beta \max\{1, \frac{2}{1+\beta}\}}$.

where we neglect prefactor constants that are independent of $t, N, B$. We see that in the rich regime, all exponents except for the model bottleneck are either the same or are improved. For *easy tasks* and super-easy tasks where $\beta > 1$, we recover the same approximate scaling laws as those computed in the linear model of Bordelon et al. (2024a) and Paquette et al. (2024). For hard tasks, $\beta < 1$, all exponents except for the model bottleneck term are improved. Below we will explain why each of these terms can experience an improvement in the $\beta < 1$ case. We exhibit a phase diagram all of the cases highlighted in equation 8, equation 9 in Figure 1.

**Accelerated Training in Rich Regime** The key distinction between our model and the random feature model ($\gamma = 0$) is the limiting gradient flow dynamics, which allow for acceleration due to feature learning. For nonzero feature learning $\gamma > 0$, our theory predicts that in the $N \to \infty$ limit, the loss scales as a power law $\mathcal{L}(t) \sim t^{-\chi(\beta)}$ where the exponent $\chi(\beta)$ satisfies the following self-consistent equation

$$\chi(\beta) = -\lim_{t \to \infty} \frac{1}{\ln t} \ln \left[ \sum_k (w_k^\star)^2 \lambda_k \exp\left(-\lambda_k \left[t + \gamma t^{2-\chi}\right]\right) \right] = \beta \max \left\{1, \frac{2}{1+\beta}\right\}. \quad (10)$$

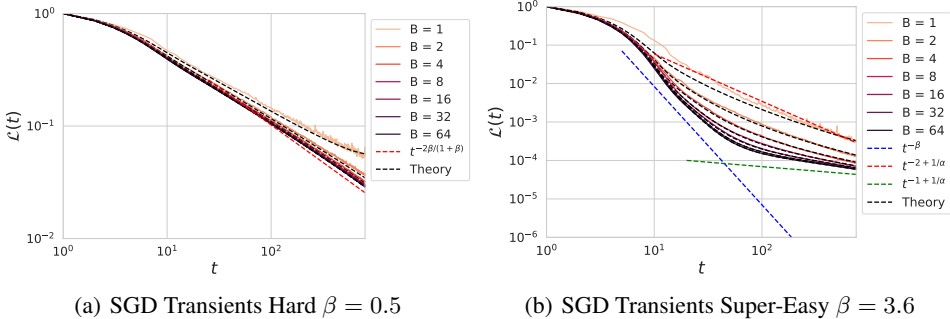

(a) SGD Transients Hard $\beta = 0.5$      (b) SGD Transients Super-Easy $\beta = 3.6$

Figure 3: SGD Transients in feature learning regime. (a) In the hard regime, the SGD noise does not significantly alter the scaling behavior, but does add some additional variance to the predictor. As $B \to \infty$, the loss converges to the $t^{-2\beta/(1+\beta)}$ scaling. (b) In the super-easy regime, the model transitions from gradient flow scaling $t^{-\beta}$ to a SGD noise limited scaling $\frac{1}{B} t^{-2+1/\alpha}$ and finally to a finite $N$ transient scaling $\frac{1}{N} t^{-1+1/\alpha}$.

We derive this equation in Appendix E. We see that if $\beta > 1$ then we have the same scaling law as a lazy learning model $\mathcal{L}(t) \sim t^{-\beta}$. However, if the task is sufficiently hard ($\beta < 1$), then the exponent is increased to $\chi = \frac{2\beta}{1+\beta} > \beta$. The time it takes to transition to the new scaling is $t \approx \gamma^{-\frac{1}{1-\chi(\beta)}}$ as we discuss in Appendix E.2.1.

This acceleration is caused by the fact that the effective dynamical kernel $K(t)$ defined by the dynamics of our features $\tilde{\psi}(x, t)$ diverges as a powerlaw $K(t) \sim t^{1-\chi}$ when $\beta < 1$ (see Appendix E). This is due to the fact that the kernel approximation at finite $\gamma$ is not stable when training on tasks out of the RKHS. As a consequence, at time $t$, the model is learning mode $k_\star(t) \sim t^{(2-\chi)/\alpha}$ which gives a loss

$$\mathcal{L}(t) \sim \sum_{k > k_\star} (w_k^\star)^2 \lambda_k \sim \gamma^{-\beta} t^{-\beta(2-\chi)} = \gamma^{-\beta} t^{-\beta \max\left\{1, \frac{2}{1+\beta}\right\}}. \tag{11}$$

While our model predicts that the scaling *exponent* only changes for hard tasks where $\beta < 1$, it also predicts an overall decrease in training loss as $\gamma$ increases for either easy or hard tasks ( Appendix E.2.1). In Figure 2 (a)-(b) we show the the $N, B \to \infty$ limit of our theory at varying values of $\gamma$. For easy tasks $\beta > 1$, the models will always follow $\mathcal{L} \sim t^{-\beta}$ at late time, but with a potentially reduced constant when $\gamma$ is large. For hard tasks (Fig. 2 (b)) the scaling exponent improves $\mathcal{L} \sim t^{-\frac{2\beta}{1+\beta}}$ for $\gamma > 0$. The full DMFT predictions in Figure 2 (c)-(d) are plotted as dashed black lines.

**Model Bottleneck Scalings** Our theory can be used to compute finite $N$ effects in the rich regime during SGD training. In this case, the dynamics smoothly transition between following the gradient descent trajectory at early time to an asymptote that depends on $N$ as $t \to \infty$. In Figure 2 (c)-(d) we illustrate these learning curves from our theory and from finite $N$ simulations, showing a good match of the theory to experiment.

We derive the asymptotic scaling of $N^{-\alpha \min\{2, \beta\}}$ in Appendix E.3. Intuitively, at finite $N$, the dynamics only depend on the filtered signal $\left(\frac{1}{N} A(0)^\top A(0)\right) w_\star$. Thus the algorithm can only estimate, at best, the top $N$ components of $w_\star$, resulting in the following $t \to \infty$ loss

$$\mathcal{L}(N) \sim \sum_{k > N} k^{-\alpha\beta - 1} \sim N^{-\alpha\beta}. \tag{12}$$

**SGD Noise Effects** The variance in the learned model predictions due to random sampling of minibatches during SGD also alters the mean field prediction of the loss. In Figure 3, we show SGD noise effects from finite batch size $B$ for hard $\beta < 1$ and super easy $\beta > 2 - 1/\alpha$ tasks.

**Compute Optimal Scaling Laws in Feature Learning Regime** At a fixed compute budget $C = Nt$, one can determine how to allocate compute towards training time $t$ and model size $N$

| Task Difficulty | Hard $\beta < 1$ | Easy $1 < \beta < 2 - 1/\alpha$ | Super-Easy $\beta > 2 - 1/\alpha$ |
|---|---|---|---|
| Lazy ($\gamma = 0$) | $\frac{\alpha\beta}{\alpha+1}$ | $\frac{\alpha\beta}{\alpha\beta+1}$ | $1 - \frac{1}{2\alpha}$ |
| Rich ($\gamma > 0$) | $\frac{2\alpha\beta}{\alpha(1+\beta)+2}$ | $\frac{\alpha\beta}{\alpha\beta+1}$ | $1 - \frac{1}{2\alpha}$ |

Table 1: Compute optimal scaling exponents $r_C$ for the loss $\mathcal{L}_\star(C) \sim C^{-r_C}$ for tasks of varying difficulty in the feature learning regime. For $\beta > 1$, the exponents coincide with the lazy model analyzed by Bordelon et al. (2024a); Paquette et al. (2024), while for hard tasks they are improved.

using our derived exponents from the previous sections. Choosing $N, t$ optimally, we derive the following compute optimal scaling laws $\mathcal{L}_\star(C)$ in the feature learning regime $\gamma > 0$. These are also summarized in Figure 1. [5]

1. Hard task regime ($\beta < 1$): the compute optimum balances the population gradient flow term $t^{-\frac{2\beta}{1+\beta}}$ and the model bottleneck $N^{-\alpha\beta}$.

2. Easy tasks ($1 < \beta < 2 - \frac{1}{\alpha}$): the compute optimum compares gradient flow term $t^{-\beta}$ to finite $N$ transient term $\frac{1}{N}t^{-1+1/\alpha}$

3. Super easy tasks ($\beta > 2 - \frac{1}{\alpha}$): compute optimum balances the finite N transient $\frac{1}{N}t^{-1+1/\alpha}$ and SGD transient terms $\frac{1}{B}t^{-2+\frac{1}{\alpha}}$.

We work out the complete compute optimal scaling laws for these three settings by imposing the constraint $C = Nt$, identifying the optimal choice of $N$ and $t$ at fixed $t$ and verifying the assumed dominant balance. We summarize the three possible compute scaling exponents in Table 1.

In Figure 4 we compare the compute optimal scaling laws in the hard and easy regimes. We show that the predicted exponents are accurate. In Figure 3 we illustrate the influence of SGD noise on the learning curve in the super easy regime and demonstrate that the large $C$ compute optimal scaling law is given by $C^{-1+\frac{1}{2\alpha}}$.

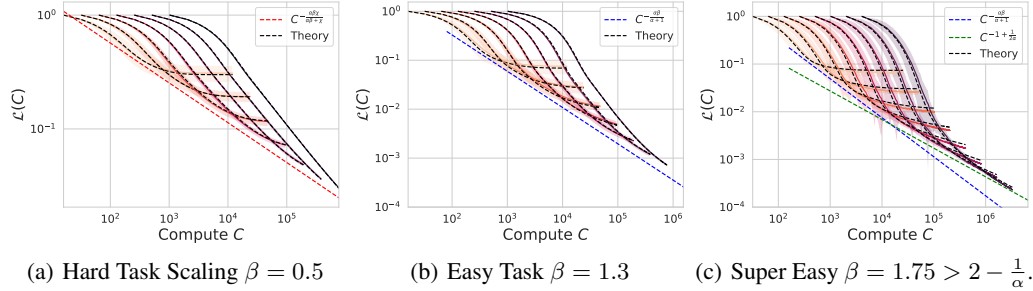

(a) Hard Task Scaling $\beta = 0.5$    (b) Easy Task $\beta = 1.3$    (c) Super Easy $\beta = 1.75 > 2 - \frac{1}{\alpha}$.

Figure 4: Compute optimal scalings in the feature learning regime ($\gamma = 0.75$). Dashed black lines are the full DMFT predictions. (a) In the $\beta < 1$ regime the compute optimal scaling law is determined by a trade-off between the bottleneck scalings for training time $t$ and model size $N$, giving $\mathcal{L}_\star(C) \sim C^{-\frac{\alpha\beta\chi}{\alpha\beta+\chi}}$ where $\chi = \frac{2\beta}{1+\beta}$ is the time-exponent for hard tasks in the rich-regime. (b) In the easy task regime $1 < \beta < 2 - \frac{1}{\alpha}$, the large $C$ scaling is determined by a competition between the bottleneck scaling in time $t$ and the leading order $1/N$ correction to the dynamics $\mathcal{L}_\star(C) \sim C^{-\frac{\alpha\beta}{\alpha+1}}$. (c) In the super-easy regime, the scaling exponent at large compute is derived by balancing the SGD noise effects with the $1/N$ transients.

## 5   EXPERIMENTS WITH DEEP NONLINEAR NEURAL NETWORKS

While our theory accurately describes simulations of our solvable model, we now aim to test if these new exponents are predictive when training deep nonlinear neural networks. Apriori, there is no reason for our toy model's predicted exponents to match those observed in deep nonlinear networks, yet in many cases they are descriptive.

---

[5]The three regimes of interest correspond to Phases I,II,III in Paquette et al. (2024). These are the only relevant regimes for trace-class $\langle K_\infty(\boldsymbol{x}, \boldsymbol{x})\rangle_{\boldsymbol{x}} = \sum_k \lambda_k < \infty$ (finite variance) kernels (equivalent to $\alpha > 1$).

### 5.1 SOBOLEV SPACES ON THE CIRCLE

We first consider training multilayer nonlinear MLPs with nonlinear activation function $\phi(h) = [\text{ReLU}(h)]^{q_\phi}$ in the mean field parameterization/$\mu$P (Geiger et al., 2020; Yang & Hu, 2021; Bordelon & Pehlevan, 2022) with dimensionless (width-independent) feature learning parameter $\gamma_0$. We consider fitting target functions $y(x)$ with $x = [\cos(\theta), \sin(\theta)]^\top$ on the circle $\theta \in [0, 2\pi]$. The eigenfunctions for randomly initialized infinite width networks are the Fourier harmonics. We consider target functions $y(\theta)$ with power-law Fourier spectra while the kernels at initialization $K(\theta, \theta')$ also admit a Fourier eigenexpansion

$$y(\theta) = \sum_{k=1}^{\infty} k^{-q} \cos(k\theta) \ , \ K(\theta, \theta') = \sum_{k=1}^{\infty} \lambda_k \cos(k(\theta - \theta')).  \tag{13}$$

We show that the eigenvalues of the kernel at initialization decay as $\lambda_k \sim k^{-2q_\phi}$ in the Appendix A. The capacity and source exponents $\alpha, \beta$ required for our theory can be computed from $q$ and $q_\phi$ as

$$\alpha = 2q_\phi \ , \ \beta = \frac{2q - 1}{2q_\phi}  \tag{14}$$

Thus task difficulty can be manipulated by altering the target function or the nonlinear activation function of the neural network. We show in Figure 5 examples of online training in this kind of network on tasks and architectures of varying $\beta$. In all cases, our theoretical prediction of $t^{-\beta \max\{1, \frac{2}{1+\beta}\}}$ provides a very accurate prediction of the scaling law.

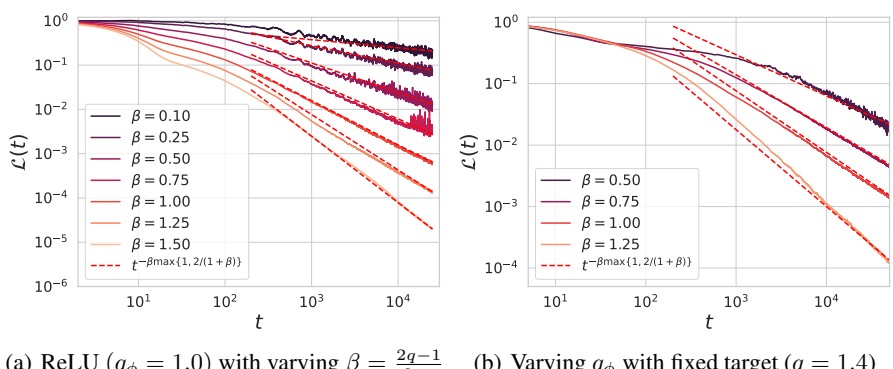

(a) ReLU ($q_\phi = 1.0$) with varying $\beta = \frac{2q-1}{2q_\phi}$    (b) Varying $q_\phi$ with fixed target ($q = 1.4$)

Figure 5: Changing the target function's Fourier spectrum or the neural network can change the scaling law in nonlinear networks trained online. These MLPs are depth 4 and width 512. (a) Our predicted exponents are compared to SGD training in a ReLU network. The exponent $\beta$ is varied by changing $q$, the decay rate for the target function's Fourier spectrum. The scaling laws are well predicted by our toy model $t^{-\beta \max\{1, \frac{2}{1+\beta}\}}$. (b) The learning exponent for a fixed target function can also be manipulated by changing properties of the model such as the activation function $q_\phi$.

### 5.2 COMPUTER VISION TASKS (MNIST AND CIFAR)

We next study networks trained on MNIST and CIFAR image recognition tasks. Our motivation is to study networks training in the online setting over several orders of magnitude in time. To this end, we adopt larger versions of these datasets: "MNIST-1M" and CIAFR-5M. We generate MNIST-1M using the denoising diffusion model (Ho et al., 2020) in Pearce (2022). We use CIFAR-5M from Nakkiran et al. (2021). Earlier results in Refinetti et al. (2023) show that networks trained on CIFAR-5M have very similar trajectories to those trained on CIFAR-10 without repetition. The resulting scaling plots are provided in Figure 6. MNIST-1M scaling is very well captured by the our theoretical scaling exponents. The CIFAR-5M scaling law exponent at large $\gamma_0$ first follows our predictions, but later enters a regime with exponent larger than what our theoretical model predicts.

## 6 DISCUSSION

We proposed a simple model of learning curves in the rich regime where the original features can evolve as a linear combination of the initial features. While the theory can give a quantitatively

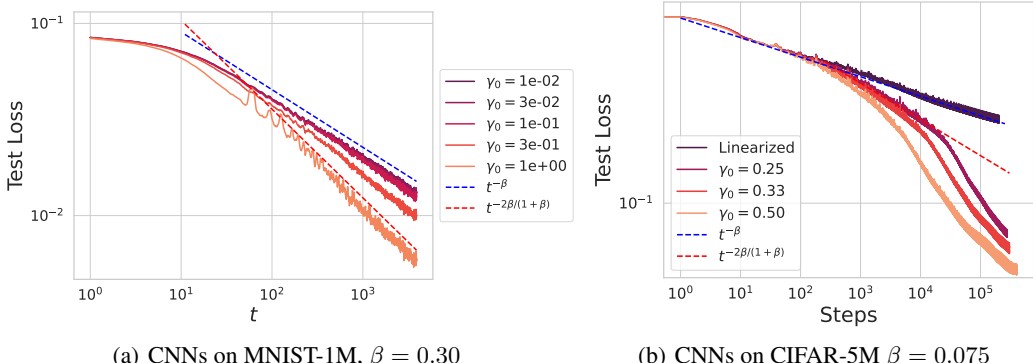

(a) CNNs on MNIST-1M, $\beta = 0.30$        (b) CNNs on CIFAR-5M $\beta = 0.075$

Figure 6: The improved scaling law with training time gives better predictions for training deep networks on real data, but still slightly underestimate improvements to the scaling law for Residual CNNs trained on CIFAR-5M, especially at large richness $\gamma_0$ (experimental details in Appendix A). (a)-(b) Training on MNIST-1M is well described by the new power law exponent from our theory. (c) CNN training on CIFAR-5M is initially well described by our new exponent, but eventually achieves a better power law.

accurate for the online learning scaling exponent in the rich regime for hard tasks, the CIFAR-5M experiment suggests that additional effects in nonlinear networks can occur after sufficient training. However, there are many weaker predictions of our theory that we suspect to hold in a wider set of settings, which we enumerate below.

**Source Hypothesis:** *Feature Learning Only Improves Scaling For $\beta < 1$.* Our model makes a general prediction that feature learning does not improve the scaling laws for tasks within the RKHS of the initial infinite width kernel. Our experiments with ReLU networks fitting functions in different Sobolev spaces with $\beta > 1$ support this hypothesis. Since many tasks using real data appear to fall outside the RKHS of the initial infinite width kernels, this hypothesis suggests that lazy learning would not be adequate to describe neural scaling laws on real data, consistent with empirical findings (Vyas et al., 2022).

**Insignificance of SGD for Hard Tasks**  Recent empirical work has found that SGD noise has little impact in online training of deep learning models (Vyas et al., 2023b; Zhao et al., 2024). Our theory suggests this may be due to the fact that SGD transients are always suppressed for realistic tasks which are often outside the RKHS of the initial kernel. The regime in which feature learning can improve the scaling law in our model is precisely the regime where SGD transients have no impact on the scaling behavior.

**Ordering of Models in Lazy Limit Preserved in Feature Learning Regime**  An additional interesting prediction of our theory is that the ordering of models by performance in the lazy regime is preserved is the same as the ordering of models in the feature learning regime. If model A outperforms model B on a task in the lazy limit ($\beta_A > \beta_B$), then model A will also perform better in the rich regime $\chi(\beta_A) > \chi(\beta_B)$ (see Figure 1). This suggests using kernel limits of neural architectures for fast initial architecture search may be viable, despite failing to capture feature learning (Park et al., 2020). This prediction deserves a greater degree of stress testing.

**Limitations and Future Directions**  There are many limitations to the current theory. First, we study mean square error loss with SGD updates, while most modern models are trained on cross-entropy loss with adaptive optimizers (Everett et al., 2024). Understanding the effect of adaptive optimizers or preconditioned updates on the scaling laws represents an important future direction. In addition, our model treats the learned features as linear combinations of the initial features, an assumption which may be violated in finite width neural networks. Lastly, while our theory is very descriptive of nonlinear networks on several tasks, we did identify a noticeable disagreement on CIFAR-5M after sufficient amounts of training. Versions of our model where the learned features are not within the span of the initial features or where the matrix $\boldsymbol{A}$ undergoes different dynamics may provide a promising avenue of future research to derive effective models of neural scaling laws.

ACKNOWLEDGEMENTS

We would like to thank Jacob Zavatone-Veth, Jascha Sohl-Dickstein, Courtney Paquette, Bruno Loureiro, and Yasaman Bahri for useful discussions. B.B. is supported by a Google PhD Fellowship. A.A. is supported by a Fannie and John Hertz Fellowship. C.P. is supported by NSF grant DMS-2134157, NSF CAREER Award IIS-2239780, and a Sloan Research Fellowship. This work has been made possible in part by a gift from the Chan Zuckerberg Initiative Foundation to establish the Kempner Institute for the Study of Natural and Artificial Intelligence.

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

APPENDIX

# A  ADDITIONAL EXPERIMENTS AND EXPERIMENTAL DETAIL

## A.1  MLPS ON SOBOLEV TASKS

The MLPs in Figure 5 were depth $L = 4$ with nonlinearities $\phi(h) = \mathrm{ReLU}(h)^{q_\phi}$, giving the following forward pass

$$f(\boldsymbol{x}) = \frac{1}{N\gamma_0}\boldsymbol{w}^3 \cdot \phi(\boldsymbol{h}^3)\ ,\ \boldsymbol{h}^{\ell+1} = \frac{1}{\sqrt{N}}\boldsymbol{W}^\ell \phi(\boldsymbol{h}^\ell)\,(\ell \in \{1,2\})\ ,\ ,\ \boldsymbol{h}^1 = \frac{1}{\sqrt{D}}\boldsymbol{W}^0\boldsymbol{x}.$$

where $D = 2$ is the input dimension. The data are sampled randomly with $\theta \sim \mathrm{Unif}[0, 2\pi]$ and are preprocesed as $\boldsymbol{x} = [\cos(\theta), \sin(\theta)]^\top \in \mathbb{R}^2$. We diagonalize neural tangent kernels (NTKs) for architectures with varying $q_\phi$ in Figure 7, showing a change in the power law spectra.

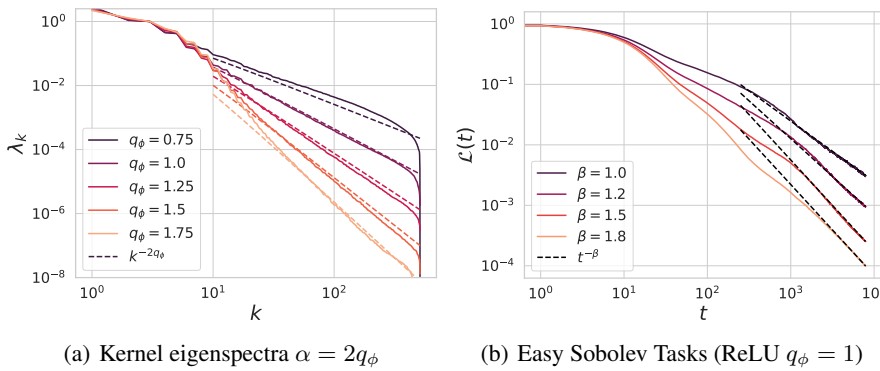

(a) Kernel eigenspectra $\alpha = 2q_\phi$  (b) Easy Sobolev Tasks (ReLU $q_\phi = 1$)

Figure 7: Additional experiments in the setting with data drawn from the circle. (a) The spectra of kernels across different nonlinearities $\phi(h) = \mathrm{ReLU}(h)^{q_\phi}$ which scale as $\lambda_k \sim k^{-2q_\phi}$. (b) More experiments in the easy task regime, show that feature learning does not alter the long time scaling behavior for $\beta > 1$.

## A.2  CNNS ON VISION TASKS

The CNN experiment on MNIST uses a depth $L = 4$ architecture with two convolutional layers and two Dense layers. The predicted exponent in the lazy regime from the spectra is $\beta = 0.3$.

For the CIFAR-5M experiment, we use the same deep residual architecture of Bordelon et al. (2024a). We reproduce the diagonalization of the kernel on CIFAR-5M in Figure 8.

## A.3  LANGUAGE MODELING TASK

We also tried an initial test of our theory in a deep transformer trained on next token prediction. In Figure 9, we plot cross entropy loss as a function of training time. Despite our theory being derived under mean square error minimization, the loss dynamics at large $\gamma$ are roughly twice the exponent as the loss dynamics at small $\gamma$.

# B  FURTHER DISCUSSION OF MODELS

We seek a model that incorporates both the bottle-necking effects of finite width observed in recent linear random feature models of scaling laws while still allowing for a notion of feature learning. Exactly solvable models of feature learning networks are relatively rare. Here, we take inspiration from the linear neural network literature Saxe et al. (2013). Linear neural networks exhibit both lazy and rich regimes of learning Woodworth et al. (2020), in which they can learn useful task-relevant features in a way that can be analytically studied Atanasov et al. (2022). In our work, we go beyond

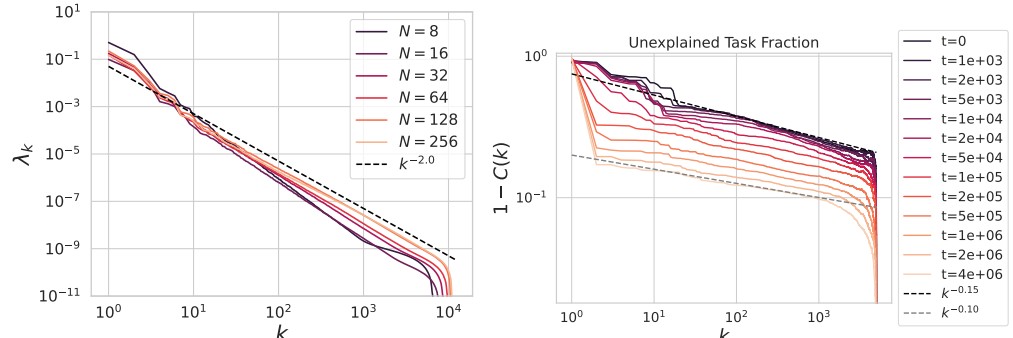

Figure 8: The spectra of ReLU residual CNNs on CIFAR-5M for varying width $N$. (a) The eigenvalues fall approximately as $\lambda_k \sim k^{-2}$ which means $\alpha \approx 2$. (b) The cumulative power spectrum $1 - C(k) = \sum_{\ell > k} (w_\ell^\star)^2 \lambda_\ell \sim k^{-0.15}$ is estimated at $t = 0$ which implies $\beta \approx 0.075$. The eigenvectors of the kernel change over time and align to the task direction, evidenced by the larger fraction of variance captured by the top eigenmode.

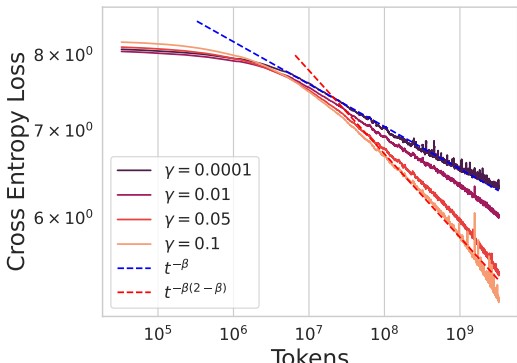

Figure 9: A depth $L = 4$ decoder-only transformer (16 heads with $d_{\text{head}} = 128$) trained on next-word prediction with SGD on the C4 dataset tokenized with the SentencePiece tokenizer. We plot cross entropy loss and fit a powerlaw $t^{-\beta}$ to lazy learning curve $\gamma = 10^{-4}$ over the interval from $10^6$ to $3 \times 10^9$ tokens. We then compute the new predicted exponent $t^{-\beta(2-\beta)}$ and compare to a simulation at $\gamma = 0.1$. Though our theoretical prediction of a doubling of the scaling exponent was derived in the context of MSE, the new scaling exponent fits the data somewhat well for this setting at early times.

these models of linear neural networks and show that linear neural networks trained on data under source and capacity conditions can improve the convergence rate compared to that predicted by kernel theory.

The model introduced in section 2 is give by a two-layer linear network acting on the $\psi_\infty(x)$:

$$f(x) = \frac{1}{N} w^\top A \psi_\infty(x). \tag{15}$$

There, we constrained it to update its weights by a form of projected gradient descent as given by Equation 4. Here, we show that running this projected gradient descent is equivalent to running ordinary gradient descent on a two layer linear network after passing $\psi_\infty$ through random projections. Define $A_0 = A(0)$ and $B(t) = A(t)A_0^+$ where + denotes the Moore-Penrose psuedoinverse. Then assuming $N < M$ and $A_0$ is rank $N$, we have

$$B(t)A_0 = A(t), \quad B(0) = I_{N \times N}. \tag{16}$$

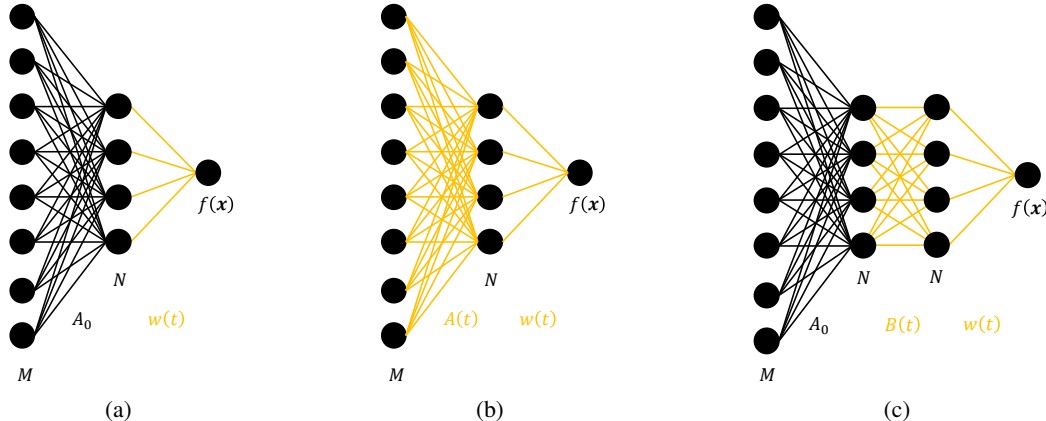

Figure 10: Three different models studied in this work and prior work. Black weights are frozen while orange weights are trainable. a) A linear random feature model with only the readout weights trainable. This model was studied in Maloney et al. (2022); Bordelon et al. (2024a); Paquette et al. (2024) as a solvable model of neural scaling laws. b) A two layer linear network with both weights trainable. This model does not incur a bottleneck due to finite width but undergoes feature learning, which improves the scaling of the loss with time. We study pure linear neural networks in Appendix G. In the main text, we train this model with a projected version of gradient descent. This is equivalent to c) and gives both finite-parameter bottlenecks as well as improvements to scaling due to feature learning.

Now consider taking $\boldsymbol{\psi}_\infty$ and passing it through $\boldsymbol{A}_0$, and then training $\boldsymbol{B}, \boldsymbol{w}$ with ordinary gradient descent. We have update equations:

$$
\boldsymbol{w}(t+1) - \boldsymbol{w}(t) = \eta \boldsymbol{B}(t) \boldsymbol{A}_0 \left( \frac{1}{B} \boldsymbol{\Psi}_\infty^\top \boldsymbol{\Psi}_\infty \right) \boldsymbol{v}^0(t)
$$
$$
\boldsymbol{B}(t+1) - \boldsymbol{B}(t) = \eta \boldsymbol{w}(t) \boldsymbol{v}^0(t)^\top \left( \frac{1}{B} \boldsymbol{\Psi}_\infty^\top \boldsymbol{\Psi}_\infty \right) \frac{1}{N} \boldsymbol{A}_0^\top
$$

(17)

Multiplying the second equation by $\boldsymbol{A}_0$ on the right recovers equation 4. Here, $\gamma$ acts as a rescaling of the learning rate for the $\boldsymbol{B}$ update equations. We illustrate this model, as well as the linear random feature and linear neural network models in Figure 10.

Several papers Maloney et al. (2022); Atanasov et al. (2023); Bordelon et al. (2024a); Atanasov et al. (2024) have studied the model given in equation 15 with frozen $\boldsymbol{A}$ under the following interpretation. The samples $\boldsymbol{\psi} \in \mathbb{R}^D$ correspond to the dataset as expressed in the space of an infinitely wide NTK at initialization. These are passed through a set of frozen random weights $\boldsymbol{W}_1$, which are thought of as the projection from the infinite width network to the finite-width empirical NTK, corresponding to a lazy network. From there, the final layer weights are not frozen and perform the analog of regression with the finite-width NTK. In Bordelon et al. (2024a), this model was shown to reproduce many of the compute optimal scaling laws observed in practice. It was also however shown there that the scaling laws for lazy networks are very different from those observed for feature-learning networks.

Our motivation is to develop a simple and solvable model of how the finite-width network features $\tilde{\boldsymbol{\psi}}(\boldsymbol{x}; t) = \boldsymbol{A}(t) \boldsymbol{\psi}_\infty(\boldsymbol{x})$ might evolve to learn useful features. The projected linear model defined above states that the $\tilde{\boldsymbol{\psi}}$ recombine themselves in such a way so that the empirical neural tangent kernel $\tilde{\boldsymbol{\psi}}(\boldsymbol{x}; t) \cdot \tilde{\boldsymbol{\psi}}(\boldsymbol{x}'; t)$ is better aligned to the task. The simple model of a linear neural network is rich enough to yield an improved power law, while still being analytically tractable.

### B.1    COMPARISON TO VERY RELEVANT PRIOR WORKS

We are using identical notation for our model's projection weights $\boldsymbol{A}$, and the error vectors $\{\boldsymbol{v}^0, \boldsymbol{v}^1, ..., \boldsymbol{v}^4\}$ to Bordelon et al. (2024a) and our dynamics match theirs in the limit of $\gamma \to 0$.

We also work with the DMFT correlation and response as in those works. Indeed, the same techniques (path integral or cavity methods) used in their prior work can be used to derive the mean field equations. Their DMFT equations could be solved exactly in Fourier space since the system was linear and time-translation-invariant (TTI). This Fourier representation is also very close to the results of Paquette et al. (2024). However, our results are significantly harder since they require tracking the evolution of $\boldsymbol{A}(t)$ and the resulting dynamics become non-TTI. However, we can still close the equations in terms of time $\times$ time matrices.

## C  DERIVATION OF THE MEAN FIELD EQUATIONS

In this setting, we derive the mean field equations for the typical test loss $\mathcal{L}(t, N, B)$ as a function of training time. To accomplish this, we have to perform disorder averages over the random matrices $\boldsymbol{A}(0)$ and $\{\boldsymbol{\Psi}(t)\}_{t=0}^{\infty}$. We start by defining the following collection of fields

$$\boldsymbol{v}^0(t) = \boldsymbol{w}^\star - \frac{1}{N}\boldsymbol{A}(t)^\top \boldsymbol{w}(t)$$

$$\boldsymbol{v}^1(t) = \boldsymbol{\Psi}(t)\boldsymbol{v}^0(t) \,, \ \boldsymbol{v}^2(t) = \frac{1}{B}\boldsymbol{\Psi}(t)^\top \boldsymbol{v}^1(t)$$

$$\boldsymbol{v}^3(t) = \boldsymbol{A}(0)\boldsymbol{v}^2(t) \,, \ \boldsymbol{v}^4(t) = \frac{1}{N}\boldsymbol{A}(0)^\top \boldsymbol{v}^3(t)$$

$$\boldsymbol{v}^w(t) = \frac{1}{N}\boldsymbol{A}(0)^\top \boldsymbol{w}(t) \tag{18}$$

From these primitive fields, we can simplify the dynamics of $\boldsymbol{A}, \boldsymbol{w}(t)$

$$\boldsymbol{A}(t) = \boldsymbol{A}(0) + \eta\gamma \sum_{s<t} \boldsymbol{w}(s)\boldsymbol{v}^4(s)^\top$$

$$\boldsymbol{w}(t+1) = \boldsymbol{w}(t) + \eta\boldsymbol{A}(t)\boldsymbol{v}^2(t)$$

$$= \boldsymbol{w}(t) + \eta\boldsymbol{v}^3(t) + \eta^2\gamma \sum_{s<t} \boldsymbol{w}(s)\left[\boldsymbol{v}^4(s)\cdot\boldsymbol{v}^2(t)\right]$$

$$= \boldsymbol{w}(t) + \eta\boldsymbol{v}^3(t) + \eta^2\gamma \sum_{s<t} \boldsymbol{w}(s)C_3(t,s) \tag{19}$$

where we introduced the correlation function $C_3(t,s) \equiv \frac{1}{N}\boldsymbol{v}^3(t)\cdot\boldsymbol{v}^3(s) = \boldsymbol{v}^2(t)\cdot\boldsymbol{v}^4(s)$. Similarly for $\boldsymbol{v}^0(t)$ and $\boldsymbol{v}^w(t)$ we have

$$\boldsymbol{v}^0(t) = \boldsymbol{w}^\star - \boldsymbol{v}^w(t) - \eta\gamma \sum_{s<t} \boldsymbol{v}^4(s)C_w(t,s)$$

$$\boldsymbol{v}^w(t+1) = \boldsymbol{v}^w(t) + \eta\boldsymbol{v}^4(t) + \eta^2\gamma \sum_{s<t} \boldsymbol{v}^w(t)C_3(t,s) \tag{20}$$

where we introduced $C_w(t,s) \equiv \frac{1}{N}\boldsymbol{w}(t)\cdot\boldsymbol{w}(s)$. We see that, conditional on the correlation function $C_3(t,s)$, the vector $\boldsymbol{v}^w(t)$ can be interpreted as a linear filtered version of $\{\boldsymbol{v}^4(s)\}_{s<t}$ and is thus redundant. In addition, we no longer have to work with the random matrix $\boldsymbol{A}(t)$ but can rather track projections of this matrix on vectors of interest. Since all dynamics only depend on the random variables $\mathcal{V} = \{\boldsymbol{v}^0, \boldsymbol{v}^1, \boldsymbol{v}^2, \boldsymbol{v}^3\}$, we therefore only need to characterize the joint distribution of these variables.

**Disorder Averages**  We now consider the averages over the random matrices which appear in the dynamics $\{\boldsymbol{\Psi}(t)\}_{t\in\mathbb{N}}$ and $\boldsymbol{A}(0)$. This can be performed with either a path integral or a cavity derivation following the techniques of Bordelon et al. (2024a). After averaging over $\{\boldsymbol{\Psi}(t)\}_{t=0}^{\infty}$, one obtains the following process for $v^1(t)$ and $v_k^2(t)$

$$v^1(t) = u^1(t) \,, \ u^1(t) \sim \mathcal{N}(0, \delta(t-s)C_0(t,t))$$

$$v_k^2(t) = u_k^2(t) + \lambda_k v_k^0(t) \,, \ u_k^2(t) \sim \mathcal{N}\left(0, B^{-1}\delta(t-s)\lambda_k C_1(t,t)\right). \tag{21}$$

where the correlation functions $C_0$ and $C_1$ have the forms

$$C_0(t,s) = \sum_k \lambda_k \left\langle v_k^0(t) v_k^0(s) \right\rangle , \; C_1(t,s) = \left\langle v^1(t) v^1(s) \right\rangle \tag{22}$$

The average over the matrix $\boldsymbol{A}(0)$ couples the dynamics for $\boldsymbol{v}^3(t), \boldsymbol{v}^4(t)$ resulting in the following

$$v^3(t) = u^3(t) + \frac{1}{N} \sum_{s<t} R_{2,4}(t,s) v^3(s) , \; u^3(t) \sim \mathcal{N}(0, C_2(t,s))$$

$$v_k^4(t) = u_k^4(t) + \sum_{s<t} R_3(t,s) v_k^2(s) , \; u_k^4 \sim \mathcal{N}\left(0, N^{-1} C_3(t,s)\right) \tag{23}$$

where

$$C_2(t,s) = \sum_k \left\langle v_k^2(t) v_k^2(s) \right\rangle , \; C_3(t,s) = \left\langle v^3(t) v^3(s) \right\rangle \tag{24}$$

Lastly, we have the following single site equation for $w(t)$ which can be used to compute $C_w(t,s)$

$$w(t+1) - w(t) = \eta v^3(t) + \eta^2 \gamma \sum_{s<t} C_3(t,s) w(s). \tag{25}$$

**Final DMFT Equations for our Model for Online SGD**   The complete governing equations for the test loss evolution after averaging over the random matrices can be obtained from the following stochastic processes which are driven by Gaussian noise sources $\{u_k^2(t), u^3(t), u_k^4(t)\}$. Letting $\langle \cdot \rangle$ represent averages over these sources of noise, the equations close as

$$
\begin{aligned}
&v_k^0(t) = w_k^\star - v_k^w(t) - \eta\gamma \sum_{s<t} C_w(t,s) v_k^4(s) \\
&v_k^w(t+1) = v_k^w(t) + \eta v_k^4(t) + \eta^2 \gamma \sum_{s<t} C_3(t,s) v_k^w(s) \\
&v_k^2(t) = u_k^2(t) + \lambda_k v_k^0(t) , \; u_k^2(t) \sim \mathcal{N}\left(0, B^{-1} \lambda_k \delta(t-s) C_0(t,t)\right) \\
&v^3(t) = u^3(t) + \frac{1}{N} \sum_{s<t} R_{2,4}(t,s) v^3(s) , \; u^3(t) \sim \mathcal{N}(0, C_2(t,s)) \\
&w(t+1) = w(t) + \eta v^3(t) + \eta^2 \gamma \sum_{s<t} C_3(t,s) w(s) \\
&v_k^4(t) = u_k^4(t) + \sum_{s<t} R_3(t,s) v_k^2(s) , \; u_k^4(t) \sim \mathcal{N}(0, N^{-1} C_3(t,s)) \\
&R_{2,4}(t,s) = \sum_k \left\langle \frac{\partial v_k^2(t)}{\partial u_k^4(s)} \right\rangle , \; R_3(t,s) = \left\langle \frac{\partial v^3(t)}{\partial u^3(s)} \right\rangle \\
&C_0(t,s) = \sum_k \lambda_k \left\langle v_k^0(t) v_k^0(s) \right\rangle , \; C_2(t,s) = \sum_k \left\langle v_k^2(t) v_k^2(s) \right\rangle
\end{aligned}
\tag{26a}
$$

**Closing the DMFT Correlation and Response as Time $\times$ Time Matrices**   We can try using these equations to write a closed form expression for $v_k^0(t)$ which determines the generalization error. First, we start by solving the equations for $\boldsymbol{v}_k^w = \mathrm{Vec}\{v_k^w(t)\}_{t\in\mathbb{N}}$. We introduce a step function matrix which is just lower triangular matrix $[\boldsymbol{\Theta}]_{t,s} = \eta\Theta(t-s)$

$$\boldsymbol{v}_k^w = [\boldsymbol{I} - \eta\gamma\boldsymbol{\Theta}\boldsymbol{C}_3]^{-1} \boldsymbol{\Theta} \boldsymbol{v}_k^4 \equiv \boldsymbol{H}_k^w \boldsymbol{v}_k^4$$

$$\boldsymbol{H}_k^w \equiv [\boldsymbol{I} - \eta\gamma\boldsymbol{\Theta}\boldsymbol{C}_3]^{-1} \boldsymbol{\Theta} \tag{27}$$

Now, combining this with the equation for $\boldsymbol{v}_k^0 = \mathrm{Vec}\{v_k^0(t)\}$ we find

$$\boldsymbol{v}_k^0 = \boldsymbol{H}_k^0 \left[ w_k^\star \boldsymbol{1} - (\boldsymbol{H}_k^w + \eta\gamma\boldsymbol{C}_w)\left(\boldsymbol{u}_k^4 + \boldsymbol{R}_3 \boldsymbol{u}_k^2\right)\right]$$

$$\boldsymbol{H}_k^0 = [\boldsymbol{I} + \lambda_k (\boldsymbol{H}_k^w + \eta\gamma\boldsymbol{C}_w) \boldsymbol{R}_3]^{-1} \tag{28}$$

The key response function is $\boldsymbol{R}_3$ with entries $[\boldsymbol{R}_3]_{t,s} = R_3(t,s)$ satisfies the following equation

$$\boldsymbol{R}_3 = \boldsymbol{I} + \frac{1}{N}\boldsymbol{R}_{2,4}\boldsymbol{R}_3 \ , \ \boldsymbol{R}_{2,4} = -\sum_{k=1}^{M}\lambda_k \boldsymbol{H}_k^0 \left(\boldsymbol{H}_k^w + \eta\gamma\boldsymbol{C}_w\right)$$

$$\implies \boldsymbol{R}_3 = \boldsymbol{I} - \frac{1}{N}\sum_{k=1}^{M}\lambda_k \boldsymbol{H}_k^0 \left(\boldsymbol{H}_k^w + \eta\gamma\boldsymbol{C}_w\right)\boldsymbol{R}_3$$

$$= \boldsymbol{I} - \frac{1}{N}\sum_{k=1}^{M}\lambda_k \left[\boldsymbol{I} + \lambda_k\left(\boldsymbol{H}_k^w + \eta\gamma\boldsymbol{C}_w\right)\boldsymbol{R}_3\right]^{-1}\left(\boldsymbol{H}_k^w + \eta\gamma\boldsymbol{C}_w\right)\boldsymbol{R}_3 \tag{29}$$

Lastly, we can compute the correlation matrix $\boldsymbol{C}_w$ from the covariance $C_3$

$$\boldsymbol{C}_w = [\boldsymbol{I} - \eta\gamma\boldsymbol{C}_3]^{-1}\boldsymbol{\Theta}\boldsymbol{C}_3\boldsymbol{\Theta}^{\top}[\boldsymbol{I} - \eta\gamma\boldsymbol{C}_3]^{-1\top} \tag{30}$$

The remaining correlation functions are defined as

$$\boldsymbol{C}_0 = \sum_k \lambda_k \boldsymbol{H}_k^0 \left[(w_k^\star)^2 \mathbf{1}\mathbf{1}^\top + (\boldsymbol{H}_k^w + \eta\gamma\boldsymbol{C}_w)\left(\frac{1}{N}\boldsymbol{C}_3 + \frac{\lambda_k}{B}\boldsymbol{R}_3\mathrm{diag}(\boldsymbol{C}_0)\boldsymbol{R}_3^\top\right)(\boldsymbol{H}_k^w + \eta\gamma\boldsymbol{C}_w)^\top\right][\boldsymbol{H}_k^0]^\top$$

$$\boldsymbol{C}_2 = \frac{1}{B}\sum_k \lambda_k \left(\boldsymbol{I} - \lambda_k \boldsymbol{H}_k^0(\boldsymbol{H}_k^w + \eta\gamma\boldsymbol{C}_w)\boldsymbol{R}_3\right)\mathrm{diag}(\boldsymbol{C}_0)\left(\boldsymbol{I} - \lambda_k \boldsymbol{H}_k^0(\boldsymbol{H}_k^w + \eta\gamma\boldsymbol{C}_w)\boldsymbol{R}_3\right)^\top$$

$$+ \sum_k \boldsymbol{H}_k^0 \left[\mathbf{1}\mathbf{1}^\top(w_k^\star)^2 + \frac{1}{N}(\boldsymbol{H}_k^w + \eta\gamma\boldsymbol{C}_w)\boldsymbol{C}_3(\boldsymbol{H}_k^w + \eta\gamma\boldsymbol{C}_w)^\top\right][\boldsymbol{H}_k^0]^\top$$

$$\boldsymbol{C}_3 = \boldsymbol{R}_3\boldsymbol{C}_2\boldsymbol{R}_3^\top \tag{31}$$

## D  OFFLINE TRAINING: TRAIN AND TEST LOSS UNDER SAMPLE REUSE

Our theory can also handle the case where samples are reused in a finite dataset $\boldsymbol{\Psi} \in \mathbb{R}^{P\times M}$. To simplify this setting we focus on the gradient flow limit (this will preserve all of the interesting finite-$P$ effects while simplifying the expressions)

$$\frac{d}{dt}\boldsymbol{w}(t) = \boldsymbol{A}(t)\left(\frac{1}{P}\boldsymbol{\Psi}^\top\boldsymbol{\Psi}\right)\boldsymbol{v}^0(t)$$

$$\frac{d}{dt}\boldsymbol{A}(t) = \gamma\,\boldsymbol{w}(t)\boldsymbol{v}^0(t)^\top\left(\frac{1}{P}\boldsymbol{\Psi}^\top\boldsymbol{\Psi}\right)\left(\frac{1}{N}\boldsymbol{A}(0)^\top\boldsymbol{A}(0)\right) \tag{32}$$

We introduce the fields

$$\boldsymbol{v}^1(t) = \boldsymbol{\Psi}\boldsymbol{v}^0(t) \ , \ \boldsymbol{v}^2(t) = \frac{1}{P}\boldsymbol{\Psi}^\top\boldsymbol{v}^1(t) \tag{33}$$

$$\boldsymbol{v}^3(t) = \boldsymbol{A}(0)\boldsymbol{v}^2(t) \ , \ \boldsymbol{v}^4(t) = \frac{1}{N}\boldsymbol{A}(0)^\top\boldsymbol{v}^3(t) \tag{34}$$

so that the dynamics can be expressed as

$$\frac{d}{dt}\boldsymbol{w}(t) = \boldsymbol{A}(t)\boldsymbol{v}^2(t)$$

$$\frac{d}{dt}\boldsymbol{A}(t) = \gamma\boldsymbol{w}(t)\boldsymbol{v}^4(t)^\top \tag{35}$$

As before we also introduce the following field which shows up in the $\boldsymbol{v}^0(t)$ dynamics

$$\boldsymbol{v}^w(t) = \frac{1}{N}\boldsymbol{A}(0)^\top\boldsymbol{w}(t) \tag{36}$$

**Data Average**  The average over the frozen data matrix $\boldsymbol{\Psi} \in \mathbb{R}^{P\times M}$

$$v_k^2(t) = u_k^2(t) + \lambda_k \int_0^t ds R_1(t,s)v_k^0(s) \ , \ u_k^2(t) \sim \mathcal{N}(0, P^{-1}\lambda_k C_1(t,s)) \tag{37}$$

$$v^1(t) = u^1(t) + \frac{1}{P}\int_0^t ds R_{0,2}(t,s)v^1(s) \ , \ u^1(t) \sim \mathcal{N}(0, C_0(t,s)) \tag{38}$$

**Feature Projection Average** Next we average over $\boldsymbol{A}(0) \in \mathbb{R}^{N \times M}$ with $N/M = \nu$ which yields

$$v_k^4(t) = u_k^4(t) + \int_0^t ds R_3(t, s) v_k^2(s) \ , \ u_k^4(t) \sim \mathcal{N}(0, N^{-1} C_3(t, s)) \tag{39}$$

$$v^3(t) = u^3(t) + \frac{1}{N} \int_0^t ds R_{2,4}(t, s) v^3(s) \ , \ u^3(t) \sim \mathcal{N}(0, C_2(t, s)) \tag{40}$$

We can now simply plug these equations into the dynamics of $\boldsymbol{w}(t), \boldsymbol{v}^w(t), \boldsymbol{v}^0(t)$ to obtain the final DMFT equations.

**Final DMFT Equations for Data Reuse Setting** The complete governing equations for the test loss evolution after averaging over the random matrices $\{\boldsymbol{\Psi}, \boldsymbol{A}\}$ can be obtained from the following stochastic processes which are driven by Gaussian noise sources $\{u_k^2(t), u^3(t), u_k^4(t)\}$. Letting $\langle \cdot \rangle$ represent averages over these sources of noise, the equations close as

$$
\begin{aligned}
&v_k^0(t) = w_k^\star - v_k^w(t) - \gamma \int_0^t ds C_w(t, s) v_k^4(s) \\[6pt]
&\partial_t v_k^w(t) = v_k^4(t) + \gamma \int_0^t ds C_3(t, s) v_k^w(s) \\[6pt]
&v^1(t) = u^1(t) + \frac{1}{P} \int_0^t ds R_{0,2}(t, s) v^1(s) \ , \ u^1(t) \sim \mathcal{N}(0, C_0(t, s)) \\[6pt]
&v_k^2(t) = u_k^2(t) + \lambda_k \int ds R_1(t, s) v_k^0(t) \ , \ u_k^2(t) \sim \mathcal{N}\left(0, P^{-1} \lambda_k C_1(t, s)\right) \\[6pt]
&v^3(t) = u^3(t) + \frac{1}{N} \sum_{s<t} R_{2,4}(t, s) v^3(s) \ , \ u^3(t) \sim \mathcal{N}(0, C_2(t, s)) \\[6pt]
&w(t+1) = w(t) + \eta v^3(t) + \eta^2 \gamma \sum_{s<t} C_3(t, s) w(s) \\[6pt]
&v_k^4(t) = u_k^4(t) + \sum_{s<t} R_3(t, s) v_k^2(s) \ , \ u_k^4(t) \sim \mathcal{N}(0, N^{-1} C_3(t, s)) \\[6pt]
&R_{0,2}(t, s) = \sum_k \lambda_k \left\langle \frac{\partial v_k^0(t)}{\partial u_k^2(s)} \right\rangle \ , \ R_1(t, s) = \left\langle \frac{\partial v^1(t)}{\partial u^1(s)} \right\rangle \\[6pt]
&R_{2,4}(t, s) = \sum_k \left\langle \frac{\partial v_k^2(t)}{\partial u_k^4(s)} \right\rangle \ , \ R_3(t, s) = \left\langle \frac{\partial v^3(t)}{\partial u^3(s)} \right\rangle \\[6pt]
&C_0(t, s) = \sum_k \lambda_k \left\langle v_k^0(t) v_k^0(s) \right\rangle \ , \ C_1(t, s) = \left\langle v^1(t) v^1(s) \right\rangle \\[6pt]
&C_2(t, s) = \sum_k \left\langle v_k^2(t) v_k^2(s) \right\rangle \ , \ C_3(t, s) = \left\langle v^3(t) v^3(s) \right\rangle
\end{aligned}
\tag{41a}
$$

The $\gamma \to 0$ limit of these equations recovers the DMFT equations from Bordelon et al. (2024a) which analyzed the random feature (static $\boldsymbol{A}$) case.

## E   BOTTLENECK SCALINGS FOR POWER LAW FEATURES

In this setting, we investigate the scaling behavior of the model under the source and capacity conditions described in the main text:

$$\lambda_k \sim k^{-\alpha} \ , \ (w_k^\star)^2 \lambda_k \sim k^{-\beta\alpha - 1} \tag{42}$$

### E.1   TIME BOTTLENECK

In this section we compute the loss dynamics in the limit of $N, B \to \infty$. We start with a perturbative argument that predicts a scaling law of the form $\mathcal{L}(t) \sim t^{-\beta(2-\beta)}$ for $\beta < 1$. We then use this

approximation to bootstrap more and more precise estimates of the exponent. The final prediction is the limit of infinitely many approximation steps which recovers $\mathcal{L}(t) \sim t^{-\frac{2\beta}{1+\beta}}$. We then provide a self-consistency derivation of this exponent to verify that it is the stable fixed point of the exponent.

### E.1.1 WARMUP: PERTURBATION EXPANSION OF THE DMFT ODER PARAMETERS

First, in this section, we investigate the $N, B \to \infty$ limit of the DMFT equations and study what happens for small but finite $\gamma$. This perturbative approximation will lead to an approximation $\mathcal{L} \sim t^{-\beta(2-\beta)}$. In later sections, we will show how to refine this approximation to arrive at our self-consistently computed exponent $\frac{2\beta}{1+\beta}$. In the $N, B \to \infty$ limit, the DMFT equations simplify to

$$\boldsymbol{R}_3 \to \boldsymbol{I} \ , \ \boldsymbol{u}_k^4 \to 0 \ , \ \boldsymbol{u}_k^2 \to 0 \ , \ \boldsymbol{v}_k^4 \to \lambda_k \boldsymbol{v}_k^0 \ , \ \boldsymbol{C}_3 \to \boldsymbol{C}_2. \tag{43}$$

The dynamics in this limit have the form

$$v_k^0(t) = w_k^\star - v_k^w(t) - \eta\gamma\lambda_k \sum_{s<t} C_w(t,s) v_k^0(s)$$

$$v_k^w(t+1) - v_k^w(t) = \eta\lambda_k v_k^0(t) + \eta\gamma \sum_{s<t} C_2(t,s) v_k^w(s)$$

$$w(t+1) - w(t) = \eta v^3(t) + \eta\gamma \sum_{s<t} C_2(t,s) w(s)$$

$$C_2(t,s) = \sum_k \lambda_k^2 \left\langle v_k^0(t) v_k^0(s) \right\rangle \ , \ C_w(t,s) = \langle w(t)w(s) \rangle \tag{44}$$

These exact dynamics can be simulated as we do in Figure 2. However, we can obtain the correct rate of convergence by studying the following Markovian continuous time approximation of the above dynamics where we neglect the extra $\mathcal{O}(\gamma)$ term in the $v_k^w(t)$ dynamics

$$\frac{d}{dt} v_k^0(t) \approx \lambda_k v_k^0(t) - \gamma\lambda_k C_w(t) v_k^0(t) \ , \ C_w(t) \equiv C_w(t,t)$$

$$\partial_t C_w(t) \approx C_2(t) + \mathcal{O}(\gamma) \ , \ C_2(t) \equiv C_2(t,t) \tag{45}$$

The solution for the error along the $k$-th eigenfunction will take the form

$$v_k^0(t) \approx \exp\left(-\lambda_k t - \gamma\lambda_k \int_0^t ds\, C_w(s)\right) w_k^\star \tag{46}$$

We next solve for the dynamics of $C_2(t)$ and $C_w(t)$ in the leading order $\gamma \to 0$ limit (under the linear dynamics)

$$C_2(t) \sim \sum_k \lambda_k^2 (w_k^\star)^2 e^{-2\lambda_k t} \sim \int dk\, k^{-\alpha-\beta\alpha-1} e^{-k^{-\alpha}t} \sim t^{-\beta}$$

$$C_w(t) \sim \begin{cases} t^{1-\beta} & \beta < 1 \\ C_w(\infty) & \beta > 1 \end{cases} \tag{47}$$

where $C_w(\infty)$ is a limiting finite value of $C_w(t)$.

$$\frac{v_k^0(t)}{w_k^\star} \approx \begin{cases} \exp\left(-\lambda_k t - \gamma\lambda_k t^{2-\beta}\right) & \beta < 1 \\ \exp\left(-\lambda_k[1 + \gamma C_w(\infty)]t\right) & \beta > 1 \end{cases} \tag{48}$$

For $\beta < 1$, the feature learning term will eventually dominate. The mode $k_\star(t)$ which is being learned at time $t$ satisfies

$$k_\star(t) \sim t^{(2-\beta)/\alpha} \tag{49}$$

which implies that the

$$\mathcal{L} \approx \sum_{k>k_\star} (w_k^\star)^2 \lambda_k \sim t^{-\beta(2-\beta)} \tag{50}$$

However, this solution actually *over-estimates* the exponent. To derive a better approximation of the exponent, we turn to a Markovian perspective on the dynamics which holds as $N, B \to \infty$.

### E.1.2 MATRIX PERTURBATION PERSPECTIVE ON THE TIME BOTTLENECK SCALING WITH MARKOVIAN DYNAMICS

The limiting dynamics in the $N, B \to \infty$ limit can also be expressed as a Markovian system in terms of the vector $\boldsymbol{v}^0(t)$ and a matrix $\boldsymbol{M}(t)$ which preconditions the gradient flow dynamics

$$
\frac{d}{dt} \boldsymbol{v}^0(t) = -\boldsymbol{M}(t)\boldsymbol{\Lambda}\boldsymbol{v}^0(t) \,, \; \boldsymbol{M}(t) = \left[ \frac{1}{N} \boldsymbol{A}(t)^\top \boldsymbol{A}(t) + \frac{\gamma}{N} |\boldsymbol{w}(t)|^2 \boldsymbol{I} \right]
$$

$$
\frac{d}{dt} \boldsymbol{M}(t) = \gamma \left( \boldsymbol{w}_\star - \boldsymbol{v}^0(t) \right) \boldsymbol{v}^0(t)^\top \boldsymbol{\Lambda} + \gamma \boldsymbol{\Lambda} \boldsymbol{v}^0(t) \left( \boldsymbol{w}_\star - \boldsymbol{v}^0(t) \right)^\top + 2\gamma (\boldsymbol{w}_\star - \boldsymbol{v}^0(t))^\top \boldsymbol{\Lambda} \boldsymbol{v}^0(t) \boldsymbol{I}
$$

$$(51)$$

We can rewrite this system in terms of the function $\boldsymbol{\Delta}(t) = \boldsymbol{\Lambda}^{1/2} \boldsymbol{v}^0(t)$ with $\boldsymbol{y} = \boldsymbol{\Lambda}^{1/2} \boldsymbol{w}_\star$ and the Hermitian kernel matrix $\boldsymbol{K} = \boldsymbol{\Lambda}^{1/2} \boldsymbol{M}(t) \boldsymbol{\Lambda}^{1/2}$ then

$$
\frac{d}{dt} \boldsymbol{\Delta}(t) = -\boldsymbol{K}(t)\boldsymbol{\Delta}(t).
$$

$$
\frac{d}{dt} \boldsymbol{K}(t) = \gamma(\boldsymbol{y} - \boldsymbol{\Delta}(t))\boldsymbol{\Delta}(t)^\top \boldsymbol{\Lambda} + \gamma \boldsymbol{\Lambda} \boldsymbol{\Delta}(t)(\boldsymbol{y} - \boldsymbol{\Delta}(t))^\top + 2\gamma(\boldsymbol{y} - \boldsymbol{\Delta}(t)) \cdot \boldsymbol{\Delta}(t) \, \boldsymbol{\Lambda} \quad (52)
$$

The test loss can be expressed as $\mathcal{L}(t) = |\boldsymbol{\Delta}(t)|^2$.

**Loss Dynamics Dominated by the Last Term in Kernel Dynamics**  We note that the loss dynamics satisfy the following dynamics at large time $t$

$$
\frac{d}{dt} \mathcal{L}(t) = -\boldsymbol{\Delta}(t)^\top \boldsymbol{K}(t)\boldsymbol{\Delta}(t)
$$

$$
= -\boldsymbol{\Delta}(t)\boldsymbol{\Lambda}\boldsymbol{\Delta}(t) - 2\gamma \int_0^t ds[(\boldsymbol{y} - \boldsymbol{\Delta}(s)) \cdot \boldsymbol{\Delta}(t)]\boldsymbol{\Delta}(t)^\top \boldsymbol{\Lambda}\boldsymbol{\Delta}(s)
$$

$$
- 2\gamma(\boldsymbol{\Delta}(t)^\top \boldsymbol{\Lambda}\boldsymbol{\Delta}(t)) \int_0^t ds(\boldsymbol{y} - \boldsymbol{\Delta}(s)) \cdot \boldsymbol{\Delta}(s)
$$

$$
\sim -\underbrace{\boldsymbol{\Delta}(t)^\top \boldsymbol{\Lambda}\boldsymbol{\Delta}(t)}_{\text{Lazy Limit}} - \underbrace{2\gamma[\boldsymbol{y} \cdot \boldsymbol{\Delta}(t)] \int_0^t ds\boldsymbol{\Delta}(t)^\top \boldsymbol{\Lambda}\boldsymbol{\Delta}(s)}_{\text{Subleading}} - \underbrace{2\gamma(\boldsymbol{\Delta}(t)^\top \boldsymbol{\Lambda}\boldsymbol{\Delta}(t)) \int_0^t ds \, \boldsymbol{y} \cdot \boldsymbol{\Delta}(s)}_{\text{Dominant}}
$$

$$(53)$$

$$
\approx -\boldsymbol{\Delta}(t)^\top \boldsymbol{\Lambda}\boldsymbol{\Delta}(t) \left[ 1 + 2\gamma \int_0^t ds\boldsymbol{y} \cdot \boldsymbol{\Delta}(s) \right]. \tag{54}
$$

One can straightforwardly verify that the middle term is subleading compared to the final term is that under the ansatz that $\Delta_k(t) \sim \exp\left(-\lambda_k t^{2-\chi}\right)$ where $\beta \leq \chi \leq 1$ for $\beta < 1$. We can therefore focus on the last term when deriving corrections to the scaling law.

**Intuition Pump: Perturbative Level 1 Approximation**  In the lazy limit $\gamma \to 0$, $\boldsymbol{K}(t) = \boldsymbol{\Lambda}$ for all $t$. However, for $\gamma > 0$ this effective kernel matrix $\boldsymbol{K}(t)$ evolves in a task-dependent manner. To compute $\boldsymbol{K}$ will approximate $\boldsymbol{M}(t)$ with its leading order dynamics in $\gamma$, which are obtained by evaluating the $\boldsymbol{v}^0(t)$ dynamics with the lazy learning $\gamma \to 0$ solution. We can thus approximate the kernel matrix $\boldsymbol{K}(t)$ dynamics as

$$
\boldsymbol{K}(t) \approx \boldsymbol{\Lambda} + \gamma \boldsymbol{y}\boldsymbol{y}^\top \left(\boldsymbol{I} - \exp\left(-\boldsymbol{\Lambda}t\right)\right)^\top + \gamma \left(\boldsymbol{I} - \exp(-\boldsymbol{\Lambda}t)\right) \boldsymbol{y}\boldsymbol{y}^\top
$$

$$
+ 2\gamma \left[ \boldsymbol{y}^\top \boldsymbol{\Lambda}^{-1} \left(\boldsymbol{I} - \exp(-\boldsymbol{\Lambda}t)\right) \boldsymbol{y} \right] \boldsymbol{\Lambda} \tag{55}
$$

From this perspective we see that the kernel has two dynamical components. First, a low rank spike grows in the kernel, eventually converging to the rank one matrix $\boldsymbol{y}\boldsymbol{y}^\top$. In addition, there is a scale growth of the existing eigenvalues due to the last term $\left[\boldsymbol{y}^\top \boldsymbol{\Lambda}^{-1} \left(\boldsymbol{I} - \exp(-\boldsymbol{\Lambda}t)\right) \boldsymbol{y}\right] \boldsymbol{\Lambda}$, which will approach the value of the RKHS norm of the target function as $t \to \infty$. The eigenvalues $\{\mathcal{K}_k(t)\}_{k=1}^\infty$ of the kernel $\boldsymbol{K}(t)$ evolve at leading order as the diagonal entries. Assuming that $\beta < 1$ these terms

increase with $t$ as

$$\mathcal{K}_k(t) \sim \lambda_k + 2\gamma y_k^2 \left(1 - e^{-\lambda_k t}\right) + 2\gamma\lambda_k \sum_\ell \frac{y_\ell^2}{\lambda_\ell}(1 - e^{-\lambda_\ell t})$$

$$\sim \lambda_k + 2\gamma y_k^2 \left(1 - e^{-\lambda_k t}\right) + 2\gamma\lambda_k t^{1-\beta} \tag{56}$$

The dynamics for the errors can be approximated as

$$\frac{\partial}{\partial t}\Delta_k(t) \sim -\mathcal{K}_k(t)\Delta_k(t)$$

$$\implies \Delta_k(t) \sim \exp\left(-\int_0^t ds\, \mathcal{K}_k(s)\right)\sqrt{\lambda_k}w_k^\star$$

$$\sim \exp\left(-\lambda_k t - 2\gamma\lambda_k(w_k^\star)^2 t - 2\gamma\lambda_k t^{2-\beta}\right)\sqrt{\lambda_k}w_k^\star \tag{57}$$

For sufficiently large $t$, the final term dominates and the mode $k_\star(t)$ which is being learned at time $t$ is

$$k_\star(t) \sim t^{\frac{2-\beta}{\alpha}} \tag{58}$$

The test loss is simply the variance in the unlearned modes

$$\mathcal{L}(t) \sim \sum_{k>k_\star} (w_k^\star)^2 \lambda_k \sim t^{-\beta(2-\beta)}. \tag{59}$$

**Bootstrapping a More Accurate Exponent** From the previous argument, we started with the lazy learning limiting dynamics for $v_k^0(t) \sim e^{-\lambda_k t} w_k^\star$ and used these dynamics to estimate the rate at which $M(t)$ (or equivalently $K(t)$) changes. This lead to an improved rate of convergence for the mode errors $v_k^0(t)$, which under this next order approximation decay as $v_k^0(t) \sim e^{-\lambda_k t^{2-\beta}} w_k^\star$. Supposing that the errors decay at this rate, we can estimate the dynamics of $M(t)$

$$\frac{d}{dt}\mathcal{K}_k(t) \approx \lambda_k \sum_\ell (w_\ell^\star)^2 \lambda_\ell e^{-\lambda_\ell t^{2-\beta}} \approx \lambda_k t^{-\beta(2-\beta)} , \tag{60}$$

$$\implies v_k^0(t) \sim \exp\left(-\lambda_k t^{2-\beta(2-\beta)}\right) w_k^\star , \text{ (Level 2 Approximation)} \tag{61}$$

We can imagine continuing this approximation scheme to higher and higher levels which will yield a series of better approximations to the power law

$$\mathcal{L}(t) \sim \begin{cases} t^{-\beta} & \text{Level 0 Approximation} \\ t^{-\beta(2-\beta)} & \text{Level 1 Approximation} \\ t^{-\beta[2-\beta(2-\beta)]} & \text{Level 2 Approximation} \\ t^{-\beta[2-\beta(2-\beta(2-\beta))]} & \text{Level 3 Approximation} \\ \dots \end{cases} \tag{62}$$

We plot the first few of these in Figure 11, showing that they approach a limit as the number of levels diverges. As $n \to \infty$, this geometric series will eventually converge to $t^{-\frac{2\beta}{1+\beta}}$.

### E.2 SELF-CONSISTENT DERIVATION OF THE INFINITE LEVEL SCALING LAW EXPONENT

From the above argument, it makes sense to wonder whether or not there exists a fixed point to this series of approximations that will actually yield the correct exponent in the limit of infinitely many steps. Indeed in this section, we find that this limit can be computed self-consistently

$$\frac{d}{dt}M(t) \approx \gamma\lambda_k \sum_\ell w_\ell^\star \lambda_\ell v_\ell^0(t) \tag{63}$$

$$\implies M(t) = 1 + \gamma \int_0^t ds \sum_\ell w_\ell^\star \lambda_\ell v_\ell^0(t) \tag{64}$$

$$v_k(t) \sim \exp\left(-\lambda_k \int_0^t dt'\, M(t')\right) w_k^\star \tag{65}$$

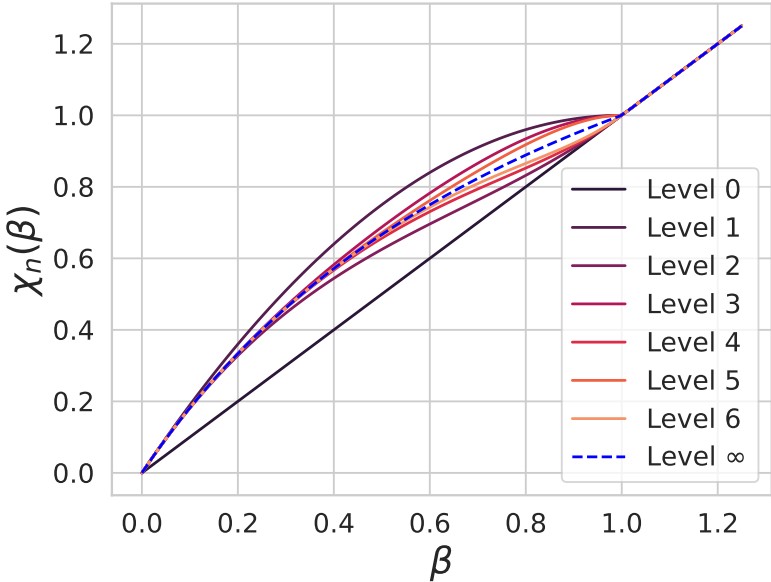

Figure 11: Predictions for the loss scaling exponent $\chi_n(\beta)$ at varying levels $n$ of the approximation scheme. Our final prediction is the infinite level limit which gives $\frac{2}{1+\beta}$. This agrees with a self-consistency argument.

We can define the intermediate variable

$$B(t) = \sum_k w_k^\star \lambda_k v_k^0(t) \tag{66}$$

which has self-consistent equation

$$B(t) = \sum_k \lambda_k (w_k^\star)^2 \exp\left(-\lambda_k \int_0^t dt' \left[1 + \gamma \int_0^{t'} ds B(s)\right]\right) \tag{67}$$

We can seek a solution of the form $B(t) \sim t^{-\chi}$. This yields

$$t^{-\chi} \approx t^{-\max\{\beta, \beta(2-\chi)\}} \implies \chi = \beta \max\left\{1, \frac{2}{\beta+1}\right\}. \tag{68}$$

Using the solution for $B(t)$, we can also derive the scaling for the loss which is identical $\mathcal{L}(t) \sim t^{-\chi}$. We note that this argument also leads to an approximate doubling of the exponent compared to the lazy case for $\beta < 1$, however this is slightly disagreement with the perturbative approach which yields $\beta(2-\beta)$ for $\beta < 1$.

This argument is where we developed the general expression that determines $\chi$ which was provided in the main text

$$\chi = -\lim_{t \to \infty} \frac{1}{\ln t} \ln\left[\sum_k (w_k^\star)^2 \lambda_k \exp\left(-\lambda_k \left[t + \gamma t^{2-\chi}\right]\right)\right]. \tag{69}$$

### E.2.1 EFFECT OF $\gamma$ ON THE SCALING LAW

Using the results of the previous sections, we see that the loss curve transitions from the lazy scaling at a time $t_{\text{transition}}$ which satisfies

$$t_{\text{transition}} \approx \gamma t_{\text{transition}}^{2-\chi} \implies t_{\text{transition}} \approx \gamma^{-\frac{1}{1-\chi}} \tag{70}$$

After this time, the loss will be dominated by the contributions from the feature learning term (which involves $\gamma$). At time $t$ the mode which is being learned is $k_\star \approx \gamma^{1/\alpha} t^{\frac{2-\chi}{\alpha}}$ The loss will scale as

$$\mathcal{L} \approx \sum_{k > k_\star} (w_k^\star)^2 \lambda_k \sim t^{-\beta(2-\chi)} \gamma^{-\beta} \tag{71}$$

This indicates that $\gamma$ modifies the prefactor but will not change the asymptotic exponent provided that it is nonzero. We summarize these results as

1. For times $t \ll \gamma^{-\frac{1}{1-\chi(\beta)}}$ the dynamics closely track the lazy learning curve.

2. At timescale $t \approx \gamma^{-\frac{1}{1-\chi(\beta)}}$ the power law transitions from the lazy learning curve to the new power law.

3. For all $t \gg \gamma^{\frac{1}{1-\chi(\beta)}}$, the loss looks like $L \approx \gamma^{-\beta} t^{-\chi(\beta)}$.

Thus our learning curve in the $N, B \to \infty$ limit has the following form.

$$\mathcal{L}(t) = \begin{cases} t^{-\beta} & t < \gamma^{-\frac{1}{1-\chi}} \\ t^{-\chi(\beta)} \gamma^{-\beta} & t > \gamma^{-\frac{1}{1-\chi}} \end{cases} \tag{72}$$

### E.3 FINITE MODEL SIZE BOTTLENECK

In the limit of $t \to \infty$, the dynamics for $\{v_k^0(t)\}$ will converge to a fixed point that depends on $N$. To ascertain the value of this fixed point, we first must compute the asymptotics. First, we note that correlation and response functions reach the following fixed point behaviors

$$\lim_{t,s \to \infty} \int_0^t dt' R_3(t', s) \sim r_3 \delta(t - s)$$

$$\lim_{t,s \to \infty} R_{2,4}(t, s) = r_{2,4} \, \Theta(t - s)$$

$$\lim_{t,s \to \infty} C_w(t, s) = c_w$$

$$\lim_{t,s \to \infty} \int_0^t dt' C_3(t', s) = 0 \tag{73}$$

which gives the following long time behavior

$$v_k^0(t) \sim w_k^\star - \int_0^t dt' u_k^4(t') - \lambda_k \int_0^t dt' \int_0^{t'} ds R_3(t', s) v_k^0(s) \tag{74}$$

$$\sim w_k^\star - \int_0^t dt' u_k^4(t') - \lambda_k r_3 v_k^0(t) \tag{75}$$

$$\implies r_{2,4} \sim -\sum_k \frac{\lambda_k}{1 + \lambda_k r_3} \tag{76}$$

Using the asymptotic relationship between $\frac{1}{N} r_3 r_{2,4} = \nu$, we arrive at the following self-consistent equation for $r_3$

$$1 = \frac{1}{N} \sum_k \frac{\lambda_k r_3}{1 + \lambda_k r_3} \tag{77}$$

For power law features this gives

$$N \approx \int dk \frac{k^{-\alpha} r_3}{k^{-\alpha} r_3 + 1} \approx [r_3]^{1/\alpha} \implies r_3 \sim N^\alpha \tag{78}$$

which recovers the correct scaling law with model size $N$.

$$\lim_{t,s \to \infty} C_0(t, s) = \sum_k \frac{\lambda_k (w_k^\star)^2}{(1 + \lambda_k N^\alpha)^2}$$

$$\approx N^{-2\alpha} \int_1^N dk \, k^{-\alpha\beta - 1 + 2\alpha} + \int_N^\infty dk k^{-\beta\alpha - 1} \sim N^{-\alpha \min\{2, \beta\}}$$

$$\tag{79a}$$

### E.4 Finite Data Bottleneck (Data Reuse Setting)

The data bottleneck when training on repeated $P$ training examples is very similar. In this case, the relevant response functions to track are $R_1(t,s)$ and $R_{0,2}(t,s)$ which have the following large time properties as $t, s \to \infty$

$$\int_0^t dt' R_1(t', s) \sim r_1 \delta(t-s)$$
$$R_{0,2}(t,s) \sim r_{0,2} \, \Theta(t-s)$$

Under this ansatz, we find the following expression for $r_1$ as $t \to \infty$

$$1 \sim \frac{1}{P} \sum_k \frac{\lambda_k r_1}{1 + \lambda_k r_1}. \tag{80}$$

Following an identical argument above we find that $r_1 \sim P^\alpha$, resulting in the following asymptotic test loss

$$
\begin{aligned}
\lim_{t,s \to \infty} C_0(t,s) &= \sum_k \frac{\lambda_k (w_k^\star)^2}{(1 + \lambda_k P^\alpha)^2} \\
&\approx P^{-2\alpha} \int_1^P dk\, k^{-\alpha\beta - 1 + 2\alpha} + \int_P^\infty dk\, k^{-\beta\alpha - 1} \sim P^{-\alpha \min\{2, \beta\}}.
\end{aligned}
$$

$$\tag{81a}$$

We see that in the offline case, this scaling law in dataset size matches the dataset scaling in the lazy regime. We specifically recover the same exponents as the $\gamma \to 0$ limit which was computed in prior works Bordelon et al. (2024a); Paquette et al. (2024).

## F Transient Dynamics

To compute the transient $1/N$ and $1/B$ effects, it suffices to compute the scaling of a response function / Volterra kernel at leading order.

### F.1 Leading Bias Correction at Finite $N$

At leading order in $1/N$ the bias corrections from finite model size can be obtained from the following leading order approximation of the response function $R_3(t,s)$

$$
\begin{aligned}
R_3(t,s) &\sim \delta(t-s) - \frac{1}{N} \sum_k \lambda_k e^{-\lambda_k (t^{\chi/\beta} - s^{\chi/\beta})} + \mathcal{O}(N^{-2}) \\
&\sim \delta(t-s) + \frac{1}{N} (t^{\chi/\beta} - s^{\chi/\beta})^{-1 + 1/\alpha}
\end{aligned}
\tag{82}
$$

where $\chi = \beta \max\left\{1, \frac{2}{1+\beta}\right\}$. Following Paquette et al. (2024), we note that the scaling of $R_3(t,s) - \delta(t-s)$ determines the scaling of the finite width transient. Thus the finite width effects can be approximated as

$$\mathcal{L}(t,N) \sim \underbrace{t^{-\beta \max\{1, \frac{2}{1+\beta}\}}}_{\text{Limiting Dynamics}} + \underbrace{N^{-\alpha \min\{2, \beta\}}}_{\text{Model Bottleneck}} + \underbrace{\frac{1}{N} t^{-(1-1/\alpha) \max\{1, \frac{2}{1+\beta}\}}}_{\text{Finite Model Transient}}. \tag{83}$$

In the case where $\beta > 1$ this agrees with the transient derived in Paquette et al. (2024). However for $\beta < 1$, the feature learning dynamics accelerate the decay rate of this term.

## F.2 Leading SGD Correction at Finite $B$

We start by computing the $N \to \infty$ limit of the SGD dynamics can be approximated as

$$C_0(t) \approx \sum_k (w_k^\star)^2 \lambda_k \exp\left(-2\lambda_k(t + \gamma t^{2+\chi})\right) + \frac{\eta}{B} \int_0^t ds K(t,s) C_0(s) \tag{84}$$

where $\chi = \beta \max\left\{1, \frac{2}{1+\beta}\right\}$ and $K(t,s)$ is the Volterra-kernel for SGD Paquette et al. (2021; 2024), which in our case takes the form

$$K(t,s) = \sum_k \lambda_k \exp\left(-2\lambda_k\left[(t + \gamma t^{2-\chi}) - (s - \gamma s^{2-\chi})\right]\right)$$

$$\sim \left(t^{\frac{\max(1,2-\chi)}{\alpha}} - s^{\frac{\max(1,2-\chi)}{\alpha}}\right)^{-(\alpha-1)} \tag{85}$$

Since the transient dynamics are again generated by the long time behavior of $K(t,0)$, we can approximate the SGD dynamics as

$$\mathcal{L}(t,B) \approx \underbrace{t^{-\beta \max\{1, \frac{2}{1+\beta}\}}}_{\text{Gradient Flow}} + \underbrace{\frac{\eta}{B} t^{-(1-1/\alpha) \max\{1, \frac{2}{1+\beta}\}}}_{\text{SGD Noise}}. \tag{86}$$

As before, the $\beta > 1$ case is consistent with the estimate for the Volterra kernel scaling in Paquette et al. (2024).

# G Linear Network Dynamics under Source and Capacity

In this section, we show how in a simple linear network, the advantage in the scaling properties of the loss due to larger $\gamma$ is evident. Here, we consider a simple model of a two-layer linear neural network trained with vanilla SGD online. We explicitly add a feature learning parameter $\gamma$. We study when this linear network can outperform the rate of $t^{-\beta}$ given by linear regression directly from input space. Despite its linearity, this setting is already rich enough to capture many of the power laws behaviors observed in realistic models.

## G.1 Model Definition

Following Chizat et al. (2019), the network function is parameterized as:

$$f(\boldsymbol{x};t) = \frac{1}{\gamma}(\tilde{f}(\boldsymbol{x};t) - \tilde{f}(\boldsymbol{x};0)), \quad \tilde{f}(\boldsymbol{x};t) = \boldsymbol{w}^\top \boldsymbol{A} \boldsymbol{x}, \tag{87}$$

We let $\boldsymbol{x} \in \mathbb{R}^M$ and take hidden layer width $N$ so that $\boldsymbol{A} \in \mathbb{R}^{N \times M}$, $\boldsymbol{w} \in \mathbb{R}^N$, as in the main text.

We train the network on power law data of the following form

$$\boldsymbol{x} \sim \mathcal{N}(0, \boldsymbol{\Lambda}), \quad y = \boldsymbol{w}^* \cdot \boldsymbol{x}. \tag{88}$$

We impose the usual source and capacity conditions on $\boldsymbol{\Lambda}$ and $\boldsymbol{w}^*$ as in equation 6.

## G.2 Improved Scalings only Below $\beta < 1$

We empirically find that when $\beta > 1$, large $\gamma$ networks do not achieve better scaling than small $\gamma$ ones. By contrast, when $\beta < 1$ we see an improvement to the loss scaling. We illustrate both of these behaviors in Figure 12. Empirically, we observe a rate of $t^{-2\beta/(1+\beta)}$ for these linear networks. This is the same improvement derived for the projected gradient descent model studied in the main text.

## G.3 Tracking the Rank One Spike

In the linear network setting, one can show that this improved scaling is due to the continued the growth of a rank one spike in the first layer weights $\boldsymbol{A}$ of the linear network. By balancing, as in Du et al. (2018), this is matched by the growth of $|\boldsymbol{w}|^2$. This which will continue to grow extensively in time $D$ only when $\beta < 1$. We illustrate these two cases in Figure 13.

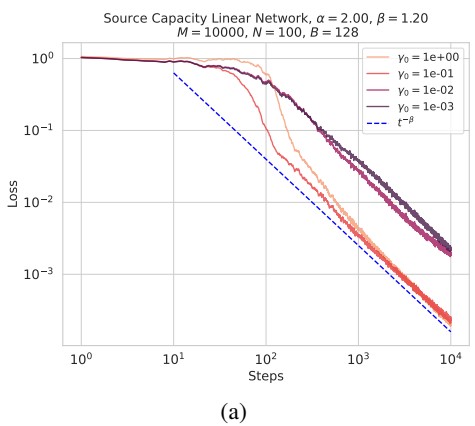 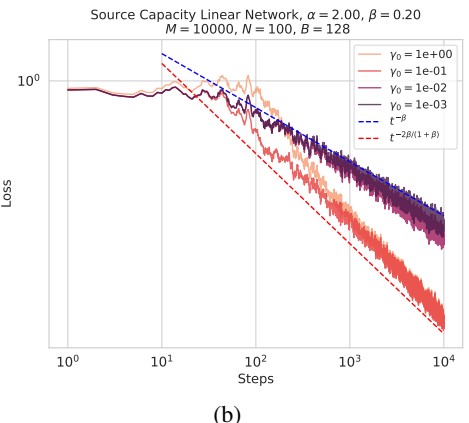

(a)                                                                 (b)

Figure 12: Linear Networks a) $\beta > 1$, where across values of $\gamma$, we observe the same asymptotic scaling going as $t^{-\beta}$ as predicted by kernel theory. b) $\beta < 1$, where feature learning linear networks achieve an improved scaling as predicted by our theory.

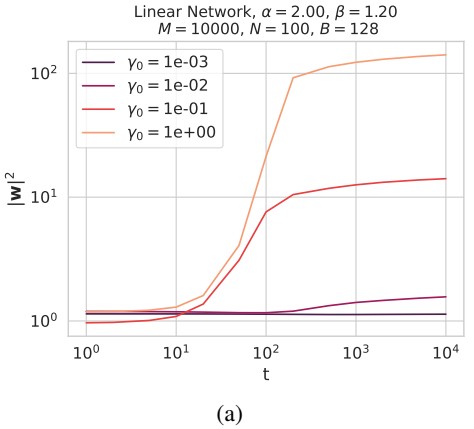 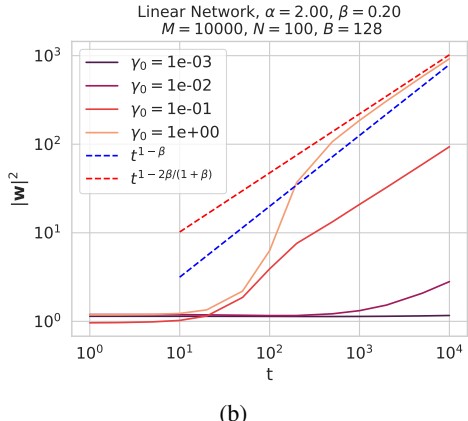

(a)                                                                 (b)

Figure 13: We study the growth of the spike as measured by $|\boldsymbol{w}|^2$. (a) For easy tasks, where $\boldsymbol{w}^*$ is finite, the spike grows to finite size, and then plateaus. This leads to a multiplicative improvement in the loss, but does not change the scaling exponent. (b) When the task is hard, $\boldsymbol{w}$ continues to grow without bound. Both the perturbative scaling of $t^{1-\beta}$ and the scaling $t^{1-2\beta(1+\beta)}$ obtained from the self-consistent equation 10 are plotted. We see excellent agreement with the latter scaling.

# H    ON THE DEFINITION OF FEATURE LEARNING

Prior works of the lazy/rich dichotomy of training neural networks define two regimes of deep network training (Chizat et al., 2019; Yang et al., 2022; Bordelon et al., 2023). The lazy regime is where the finite with empirical neural tangent kernel (eNTK) does not change (or changes negligibly) over the course of training. The rich or feature learning regime is what results when the network is trained beyond the lazy learning limit.

**Definition A** A feature learning network is one that is not in the lazy regime. That is, its eNTK changes noticeably over the course of training.

One might want to give a more stringent definition, as the above definition does not answer whether the network has learned "useful features". In general, characterizing and comparing the learned features of a deep network is a wide open research problem (Kornblith et al., 2019). However, several works (Baratin et al., 2021; Fort et al., 2020; Atanasov et al., 2022) have consistently observed

that the kernel aligns itself to relevant task directions. This is generally measured by the kernel alignment, given by a cosine-similarity $A \equiv \frac{\boldsymbol{y}^T \boldsymbol{K} \boldsymbol{y}}{\|\boldsymbol{K}\| \|\boldsymbol{y}\|^2}$, or alternatively in terms of the decomposition of the task vector $\boldsymbol{y}$ in the eigenbasis of the evolving kernel (see Figure 8 b) (Canatar & Pehlevan, 2022). This motivates a more stingent definition of feature learning which reflects **task-relevant adaptation of the kernel**.

**Definition B**    A network is said to learn useful features if the kernel-task alignment improves over its initial value at the start of training.

In our model, the networks at $\gamma > 0$ satisfy both Definition A and Definition B. The networks at $\gamma = 0$ satisfy neither. We note that neither A or B are sufficient to see an improved scaling law, and that one also requires the task to be "hard" in the linear network setting.

