# OpenReview forum: "How Feature Learning Can Improve Neural Scaling Laws"
_ICLR.cc/2025/Conference — ICLR 2025 Spotlight_

### Official Review · Reviewer_YcZy · 2024-10-29

**Soundness:** 2
**Presentation:** 3
**Contribution:** 3
**Rating:** 8
**Confidence:** 3

**Summary:**

This paper studies a minimal model for scaling laws in generalization error under feature learning non-linear neural networks. This model induces improved scaling exponents compared to kernel regression through feature learning (only) for target functions outside of the initial NTK’s RKHS. Consequently, different scaling rules of parameters and training time are asymptotically compute-optimal. These scaling rules are empirically verified in toy settings.

**Strengths:**

The paper is very well written and easy to read. To the best of my knowledge, this is the first paper that theoretically explains compute-optimal scaling laws in non-linear feature learning networks beyond the lazy regime. Separating the four bottleneck mechanisms in eq. (9) and achieving improved scaling laws through feature learning are particularly interesting accomplishments.

Overall, although this paper only studies a toy learning algorithm, I believe it makes a valuable contribution towards a theory of scaling laws for finite, feature learning neural networks, which is why I recommend acceptance.

**Weaknesses:**

Besides partially critical questions below, I consider the following points to be the main weaknesses of this paper.

- **Artificial learning setup.** The projected gradient descent with frozen first-layer weights is a constructed training procedure. This may make the analysis easier, but it remains unclear how well the results are transferable to more practical optimization procedures like SGD or Adam on **all** weights. Do you think your results can be generalized to realistic training? Do you train all weights in Section 5? This is not explicitly stated.
- **Oversimplifications.** The simplifications that make the paper very easy to read sometimes lead to oversimplifications and imprecision. For example, in l. 191, what is $N,B\gg 1$ formally supposed to mean? Overall, the approach taken in the paper seems very heuristic. For example, starting with finite $M$ and taking the limit $M\to\infty$ omits many mathematical technicalities, why the claimed limiting properties and statements are well-defined and hold.
- A small limitation is that long training times are not well covered (Figure 6b). But this is acceptable as the long training regime is, in general, still very open.

**Typos in lines:** 128, 201, 202, 476.

**Questions:**

Partially critical questions ordered from most to least important:

1. **Large feature learning regime.** Which $\gamma_0$ is optimal and is this another hyperparameter that has to be tuned, or are there useful heuristics for choosing it? In your experiments it looks like the larger the better, but the $\gamma_0$ you consider are still pretty small. I am guessing in Figure 6 (b), larger values of $\gamma_0$ would result in a better than predicted scaling exponent even earlier in training? If so, this would be an acceptable but important limitation of the studied model, which should be acknowledged. I suggest running the experiment for larger $\gamma_0$ and discussing this limitation more deeply.
2. **Importance of constants.** While reporting the pure scalings is nice for readability, in practice, how important/huge are the omitted constants? This could arbitrarily change the global optimum of the test error given finite resources. Towards answering this question of how practically useful the results are, I suggest an experiment that compares the predicted compute-optimal scaling to empirical loss measurements like in Figure 4 but for the more practical datasets in Figure 6.
3. **Error of the approximation.** How good is your approximation in eq. (9)? In which cases does it fail?
4. **Extension to label noise.** How would label noise affect the results?
5. **Long training regime.** What is happening under long training in Figure 6b? Do you have an informed guess? Are structures in the images learned that are not covered by your simple distributional assumption?
6. In eq. (6), you state $\lambda_k\sim k^{-\alpha}$ and $\lambda_k^{-\beta}\sim k^{-\alpha \beta}$. Is this a typo?
7. In l. 192, ‘the components of $\omega^*, v^0(t)$ in the k-th eigenspace of $\Lambda$’, but $\Lambda$ was defined as a diagonal matrix in eq. (2). I guess here the k-th eigenspace of $K_\infty$ is meant?

---

> ### Author Response · Authors · 2024-11-20
> **Response part 1**
>
> ### Strengths
>
> *The paper is very well written and easy to read. To the best of my knowledge, this is the first paper that theoretically explains compute-optimal scaling laws in non-linear feature learning networks beyond the lazy regime. Separating the four bottleneck mechanisms in eq. (9) and achieving improved scaling laws through feature learning are particularly interesting accomplishments.*
>
>
>
> *Overall, although this paper only studies a toy learning algorithm, I believe it makes a valuable contribution towards a theory of scaling laws for finite, feature learning neural networks, which is why I recommend acceptance.*
>
> We thank the author for appreciating our novel theoretical contributions, despite the limitations of our setting.
>
> ### Weaknesses
>
>
> **Artificial learning setup** *The projected gradient descent with frozen first-layer weights is a constructed training procedure. This may make the analysis easier, but it remains unclear how well the results are transferable to more practical optimization procedures like SGD or Adam on all weights. Do you think your results can be generalized to realistic training? Do you train all weights in Section 5? This is not explicitly stated.*
>
> We thank the reviewer for this question and for giving us an opportunity to clarify how our model relates to nonlinear networks. We emphasize that the deep networks we train in section 5 are all trained with **every layer updated with SGD with the same learning rate**. While our results do not currently try to model training with Adam, we do think capturing training a full nonlinear network with SGD is an advance in realism compared to existing theoretical models.
>
> To see how our model connects to **training all layers of a nonlinear network with SGD** we first consider the lazy limit of training a deep network $\gamma \to 0$. This is the model of [Bordelon et al 2024](https://arxiv.org/abs/2402.01092).
> 1. In this case, we start with a deep nonlinear network, whose dynamics are governed by its initial (and static) NTK.
> 2. We start by diagonalizing the infinite width NTK for this architecture which gives a collection of orthogonal eigenfunctions and eigenvalues $\lambda_k$.
> 3. The finite width kernel eigenfunctions are expanded in the infinite width basis which gives the coefficients $A$ (which will be static over training).
> 4. The learnable coefficients $w$ are updated with SGD, which will exactly mimic the behavior of the *full network will all layers trained in the lazy regime*
>
> Now, for the rich regime, the empirical kernel for a deep network evolves over training, which necessitates a dynamical system for $A$. We pick the simplest update rule for $A(t)$ that makes sense with respect to dynamics and model scaling laws (projected-SGD) and recovers the lazy limit as $\gamma \to 0$.
>
> **Oversimplifications.** *The simplifications that make the paper very easy to read sometimes lead to oversimplifications and imprecision. For example, in l. 191, what is $N,B \gg 1$ formally supposed to mean?*
>
> We thank the reviewer for pointing this out. We will try to provide links to additional detailed explanations in the Appendix. We aimed to make the main text of the paper less technical and more accessible while still providing a high level idea of the results.
>
> On the questions of $N,B \gg 1$, what we mean by this is that, while the theory accurately captures the expected loss dynamics, the variability of a randomly drawn system at size $N$ with batchsize $B$ also contains fluctuations around our loss predictions (see the errorbars on Fig 2 c-d or Fig 4). As $N,B$ increase, randomly drawn systems converge to our mean field predictions (dashed black lines). Theoretically, the dynamics of this errorbar in the loss curves could also be computed with additional effort using techniques like those employed [here](https://arxiv.org/abs/2112.05589) or [here](https://arxiv.org/abs/2304.03408) which compute fluctuations around a mean field theory.
>
>
> To address this, we have updated the second footnote on page 4 which now reads as
>
> "There are finite size fluctuations around the mean-field at small $N,B$ which are visible in errorbars in Figure 2 c-d, which could also be extracted from the theory. Alternatively, we can operate in a proportional limit with $N/M, B/M$ approaching constants, which is exact with no finite size fluctuations."

---

> > ### Author Response · Authors · 2024-11-20
> > **Response part 2**
> >
> > *A small limitation is that long training times are not well covered (Figure 6b). But this is acceptable as the long training regime is, in general, still very open.*
> >
> > Yes, our theory has not fully captured deep networks in all regimes or on all datasets. More work needs to be done to identify the scaling laws in settings like this CIFAR-5M example, which we acknowledge as a limitation of the current results.
> >
> >
> >
> > **Large feature learning regime.** *Which $\gamma_0$ is optimal and is this another hyperparameter that has to be tuned, or are there useful heuristics for choosing it? In your experiments it looks like the larger the better, but the $\gamma_0$ you consider are still pretty small. I am guessing in Figure 6 (b), larger values of $\gamma_0$ would result in a better than predicted scaling exponent even earlier in training? If so, this would be an acceptable but important limitation of the studied model, which should be acknowledged. I suggest running the experiment for larger
> >  and discussing this limitation more deeply.*
> >
> >
> > This is a good question. Under gradient flow, our theory would predict that the loss would monotonicaly improve with increasing $\gamma_0$ (as a multiplicative constant) however **the asymptotic scaling exponent does not depend on the precise value of $\gamma_0$, provided that it is nonzero**. However, in discrete time, the system will become unstable if this parameter increases too much without downscaling the learning rate, which would implicitly set an optimal value of $\gamma_0$ at the edge of stability. These discrete time effects are potentially interesting and would also be captured in our discrete time DMFT dynamics but we do not focus on them here. In our experiments we kept the raw learning rate fixed and increased $\gamma_0$ from zero to a value that kept the dynamics stable.
> >
> >
> > **Importance of constants.**  *While reporting the pure scalings is nice for readability, in practice, how important/huge are the omitted constants? This could arbitrarily change the global optimum of the test error given finite resources. Towards answering this question of how practically useful the results are, I suggest an experiment that compares the predicted compute-optimal scaling to empirical loss measurements like in Figure 4 but for the more practical datasets in Figure 6.*
> >
> > We thank the reviewer for this question and provide a few comments on the issue of prefactor constants in the scaling law.
> >
> > 1. First, we stress that we have a **full theory that captures the precise learning curve trajectories** at any time, width or batchsize $\mathcal{L}(t, N,B)$ (see dashed black lines in Fig 2, Fig 4). These equations contain all the information including constants, and exponents. Equation 9 is a coarse approximation to these full dynamics.
> > 2. From partial limits to this full theory (for instance $N \to \infty$ with $t$ fixed, or $t \to \infty$ with $N$ fixed, etc), we focused on extracting various **exponents** of the loss curve from different effects in the theory. However, this analysis could also keep track of prefactor constants with slightly more effort.
> > 3. The asymptotic compute optimal scaling law **exponent** will only depend on the **exponents** for the dominant terms, which is why we focus on the exponents.
> >
> > We agree that compute optimal scaling experiments on realistic datasets where $N$ is varied at fixed $\gamma$ are also of interest so we are hoping to add more experiments of this kind for MNIST and CIFAR.
> >
> >
> > **Error of the approximation.**  *How good is your approximation in eq. (9)? In which cases does it fail?*
> >
> > This approximation is somewhat coarse since it combines contributions computed in different partial limits of the full theory (we expect this is generally true of Chichilla-like approximations to the full loss curve in real settings as well). For instance, the first term represents the gradient descent dynamics at $N \to \infty$ while the model-bottleneck term represents the $t \to \infty$ limit. However, we show that for the purposes of computing compute-optimal scaling laws (such as those reported in the Chinchilla paper) Eq 9 does a good job. We chose to include the leading $1/N$ finite model correction to the dynamics so we could capture compute optimal scaling for easy-tasks (Figure 4 b) and the SGD transient so we could characterize super-easy tasks (Figure 4 c).
> >
> > We note that the tradeoffs for easy and super easy tasks agree with those obtained in the **linear model / lazy limit** in the work of [Paquette et al 2024](https://arxiv.org/abs/2405.15074). This is because feature learning only alters our dynamical scaling laws in the hard task regime.

---

> ### Author Response · Authors · 2024-11-20
> **Response Part 3**
>
> **What is happening under long training in Figure 6b?** *Do you have an informed guess? Are structures in the images learned that are not covered by your simple distributional assumption?*
>
> We are not entirely sure what causes the scaling behavior to change so dramatically on CIFAR-5M. One hypothesis could be that the covariance structure of the task in the limiting kernel eigenspace actually does not follow a perfect power law forever, but has a big gap that cuts off after some large but finite number of dimensions (this could be an artifact of the finite sized generative model that produced this dataset from the original finite CIFAR-10 dataset). The reason we see this in the rich networks is that they optimize over this top set of dimensions more quickly than the lazy model. In principle, we could verify this hypothesis by training a linear/lazy model for **much longer** on this distribution, but then we could potentially run into overfitting issues related to using a large number of passes over the dataset. This is currently speculation, but we hope to explore what is happening in real and synthetic datasets (like CIFAR-5M) further in future work.
>
>
> *In eq. (6), you state $\lambda_k \sim k^{-\alpha}$ and $\lambda_k^{-\beta} \sim k^{-\alpha\beta}$. Is this a typo?*
>
> Yes this is is a typo. We removed the $\lambda_k^{-\beta}$ from this line as it is completely unncessary.
>
>
> **Line 192 , eigenspace**
>
> Yes, we will correct this.

---

> > ### Comment · Reviewer_YcZy · 2024-11-20
> >
> > I thank the authors for the detailed responses. They have resolved most of my concerns. The remaining limitations, most importantly a theory for SGD and Adam on all weights in the egde-of-stability regime, can be dealt with in future work. As is, I believe the paper makes a valuable contribution and should be accepted. I updated my score accordingly.

---

### Official Review · Reviewer_QCyn · 2024-10-31

**Soundness:** 4
**Presentation:** 3
**Contribution:** 3
**Rating:** 8
**Confidence:** 3

**Summary:**

The paper studies scaling laws of the training loss with respect to relevant parameters (training time, model size and batch size), with a focus on online training, where dataset size is assumed to be unlimited.
They study training dynamics through the lens of the eigenspace decomposition of the NTK. Such eigenspace is used both to (1) formally define capacity and source of model and task and (2) to express the loss behaviour explicitly, both in the lazy learning regime ($\gamma=0$) and in the feature learning regime ($\gamma>0$), the former can be solved explicitly while the second is approximated with DMFT. Such analysis reveals two very different regimes depending on the value of the source, specifically in the hard regime (source<1) they prove better scaling laws than state of the art. Such scaling laws are studied and discussed in each component. Empirically are shown to be tight on a variety of task, including synthetic and MNIST vision, but they are shown to no be tight in the CIFAR case.

**Strengths:**

The paper is very well written and easy to follow, overall very pleasant to read.
Settings and limitations are well clarified: the authors are very open about the limits of their theory and the non-success of the CIFAR experiment. This behaviour is super valuable and very helpful for future works and possible improvements.
Derivations are explained both intuitively and formally: key concepts are present in the main paper with a good trade-off between formalism and explanation; technical details not directly useful to understand the concepts are properly deferred to the appendix.
Results are clear and well discussed: for example the scaling laws are divided in several components, each of them is further discussed and empirically studied with ad-hoc experiments.

**Weaknesses:**

The paper is overall solid and I don't see major weakness.

A big assumption made is the online setting, and the extension to finite dataset setting in Appendix D is somehow limited. However such kind of limitation has to be expected from a theoretical paper. Of course a general case analysis would be preferable, but that would very ambitious and I don't think this assumption can be considered a major weakness.

The dynamics mean field theory is not at all explained in the main paper, understandably. An informal explanation attempt is done in Line 202-211. While I consider this attempt valuable, the Equation (in line 206) is too much out of context, the notation is not defined and it ends up being practically useless for the reader. I recommend either removing (and using that space to expand the intuition informal explanation) or extending it, explaining the terms involved.

Minor detail. In Line 133 "we then diagonalize..." can be misinterpreted as "we discard the non-diagonal elements". I think overall the message is clear because right after (in Line 134) you refer to eigenfunctions, but I'd recommend trying to reformulate this sentence to avoid ambiguities.

**Questions:**

In line 160 the "allow" pops a bit out of nowhere and it hard to grasp. Following with Lines 175-176, my understanding is that you choose an arbitrary kind of feature learning dynamics, with the implicit assumption that SDG is sufficient (but not necessary) to obtain such dynamics. Being sufficient means that the scaling law you derive is at least an upper bound on the loss. While not being necessary means that a different choice of modelling dynamics for $A(t)$ may lead to tighter scaling low (which may for example explain the CIFAR experiment). Is this understanding correct? Is such implicit assumption true? Can you please argument a bit more over this choice of feature learning dynamics?

The reference "Bordelon et al. (2024a)" appears very often in the paper. It would be nice to have a short paragraph (maybe in the appendix) to clarify the common point with such paper and the novel contribution of this one over that one. Can you reply with a draft of such paragraph?

---

> ### Author Response · Authors · 2024-11-20
> **Response**
>
> We thank the reviewer for their careful reading and supportive review.
>
> ### Strengths
>
> *The paper is very well written and easy to follow, overall very pleasant to read. Settings and limitations are well clarified: the authors are very open about the limits of their theory and the non-success of the CIFAR experiment. This behaviour is super valuable and very helpful for future works and possible improvements. Derivations are explained both intuitively and formally: key concepts are present in the main paper with a good trade-off between formalism and explanation; technical details not directly useful to understand the concepts are properly deferred to the appendix. Results are clear and well discussed: for example the scaling laws are divided in several components, each of them is further discussed and empirically studied with ad-hoc experiments.*
>
> We thank the reviewer for appreciating our attempts to make the paper readable and clarifying the limitations of the current theory so that future research can improve on it.
>
>
> ### Weaknesses
>
> *A big assumption made is the online setting, and the extension to finite dataset setting in Appendix D is somehow limited. However such kind of limitation has to be expected from a theoretical paper. Of course a general case analysis would be preferable, but that would very ambitious and I don't think this assumption can be considered a major weakness.*
>
> While we do focus on the online setting and perform our simulations in this regime, our theory in Appendix D would capture the finite dataset corrections. The $\gamma \to 0$ limit of these were shown to work in Bordelon et al 2024 [dynamical scaling law paper](https://arxiv.org/abs/2402.01092) (see Figure 2d). While we haven't written a solver for the $\gamma > 0$ data repeat DMFT, we expect the theory to continue to hold under finite data. We try to extract from our theory the data scaling exponent in Appendix E.4. With more time, we could try adding finite data simulations to this paper.
>
>
>
> *The dynamics mean field theory is not at all explained in the main paper, understandably. An informal explanation attempt is done in Line 202-211. While I consider this attempt valuable, the Equation (in line 206) is too much out of context, the notation is not defined and it ends up being practically useless for the reader. I recommend either removing (and using that space to expand the intuition informal explanation) or extending it, explaining the terms involved.*
>
> Thank you for this suggestion. While we tried including a bit of detail on what the relevant correlation and response functions are for the reader familiar with DMFT methods, we realize that this may be an unhelpful distraction for readers who are unfamiliar with these techniques. We will therefore remove some unncessary detail from this section to it simpler to parse.
>
> *Minor detail. In Line 133 "we then diagonalize..." can be misinterpreted as "we discard the non-diagonal elements". I think overall the message is clear because right after (in Line 134) you refer to eigenfunctions, but I'd recommend trying to reformulate this sentence to avoid ambiguities.*
>
> Thanks for this suggestion! We will stress that we are changing coordinates to the eigenbasis of the kernel.

---

### Official Review · Reviewer_q6mk · 2024-11-04

**Soundness:** 3
**Presentation:** 3
**Contribution:** 3
**Rating:** 6
**Confidence:** 3

**Summary:**

This paper studies the scaling laws of neural networks in two regimes: lazy learning and feature learning. Eq (8) and (9) describe the power-law variation of the loss with training time \( t \), number of parameters \( N \), and batch size \( B \) in these two regimes which show that feature learning accelerates the convergence of hard tasks.

**Strengths:**

1. This paper constructs a comprehensive scaling law framework, reveals the power law relationship between loss and the number of parameters, training time, and batch size, and provides a theoretical basis for understanding the performance of neural networks.

2. This paper conducts a series of experiments to verify the theoretical results, making the conclusions of this paper more convincing.

**Weaknesses:**

1. This paper ignores the impact of gamma on the feature learning scaling law.

2. The paper lacks an analysis of the critical state between the two regimes.

3. (Minor) The colors of lines in the figures are not differentiated enough.

**Questions:**

From the experimental results in Figures 9 and 12, we can see that the scaling law does not change suddenly at $\gamma = 0$, but changes gradually at a very small value of gamma. Can this paper's analysis cover this aspect?

---

> ### Author Response · Authors · 2024-11-20
> **Response**
>
> ### Strenghts
>
> *This paper constructs a comprehensive scaling law framework, reveals the power law relationship between loss and the number of parameters, training time, and batch size, and provides a theoretical basis for understanding the performance of neural networks.*
>
>
> *This paper conducts a series of experiments to verify the theoretical results, making the conclusions of this paper more convincing.*
>
> Thank you for appreciating our paper's theoretical and experimental contributions.
>
> ### Weaknesses
>
> *This paper ignores the impact of gamma on the feature learning scaling law.*
>
> We actually can characterize the effect of $\gamma$ and will write more about the role of this parameter. While this parameter does not alter the scaling law exponent (and therefore does not alter compute optimal exponents either), it does change the **timescale where the loss curve significantly separates from the lazy learning curve**. The time when the loss curve transitions from the lazy power law to the new power law is $t\approx \gamma^{- \frac{1}{1- \chi(\beta)}}$ where $\chi = \beta\max\{1,2/(1+\beta)\}$. We note that $\chi < 1$ in the case where feature learning changes the exponent. Therefore, very small $\gamma$ requires a long time to see the transition to the new power law whereas larger $\gamma$ causes an almost immediate transition. At late time, the loss is effectively lower by a multiplicative constant $\gamma^{-\beta}$, ie the loss scales with $t$ and $\gamma$ as $\mathcal L \sim t^{-\chi(\beta)} \gamma^{-\beta}$.
>
> *The paper lacks an analysis of the critical state between the two regimes.*
>
> Our full theory provides the **complete learning curve** in terms of $\gamma$ and $t$. We approximate this full expression to get simplied expressions for the exponents (eg in equations 9 and 10). We will comment on the following facts for the hard task case. Approximating our full theory, we can provide a more detailed description of the transition that we will comment on in the paper.
>
> 1. For times $t \ll \gamma^{-\frac{1}{1 - \chi(\beta) }}$ the dynamics closely track the lazy learning curve.
> 2. At timescale $t \approx \gamma^{-\frac{1}{1- \chi(\beta)}}$ the power law transitions from the lazy learning curve to the new power law.
> 3. For all $t \gg \gamma^{\frac{1}{1-\chi(\beta)}}$, the loss looks like $L \approx \gamma^{-\beta} t^{-\chi(\beta)}$.
>
> Thus we could write something like the following
>
> \begin{align}
>     \mathcal L(t) = \begin{cases}
>     t^{-\beta}  & t < \gamma^{- \frac{1}{\chi - 1}}
>     \\
>     t^{-\chi(\beta)} \gamma^{-\beta}  & t > \gamma^{- \frac{1}{\chi - 1}}
>     \end{cases}
> \end{align}
>
> We provide this information in a new Appendix section E.2.1 and also reference some of this information in the "Accelerated Training" section.
>
> *(Minor) The colors of lines in the figures are not differentiated enough.*
>
> Thanks for this comment. For the final version, we will try to improve the color scheme to make things more differentiated.
>
>
> *From the experimental results in Figures 9 and 12, we can see that the scaling law does not change suddenly at, but changes gradually at a very small value of gamma. Can this paper's analysis cover this aspect?*
>
> Yes, we can cover this aspect. Our theory predicts that the loss will noticeably separate from the lazy learning curve at a timescale $t = \gamma^{ - \frac{1}{1 - \chi} }$.

---

> > ### Author Response · Authors · 2024-11-27
> > **Follow up**
> >
> > We are just following up to see if this reviewer has any additional questions or comments before the discussion period ends.

---

### Official Review · Reviewer_tuhi · 2024-11-05

**Soundness:** 2
**Presentation:** 1
**Contribution:** 2
**Rating:** 6
**Confidence:** 3

**Summary:**

The paper introduces a solvable two-layer linear model trained using a variant of stochastic gradient descent (SGD) to predict the scaling laws beyond the kernel (lazy training) regime. The authors identify three scaling regimes—hard, easy, and super easy tasks—under which feature learning can improve the scaling of the loss with respect to training time and compute. The paper also provides experimental validation of these theoretical predictions by training deep, nonlinear models to demonstrate the applicability of the scaling laws.

**Strengths:**

1. Tractable Theoretical Setting: The paper presents a solvable theoretical model, which allows for potentially clearer and rigorous exploration of neural scaling laws beyond the kernel regime. This tractable setting potentially makes it easier to understand and predict the dynamics of feature learning in neural networks.

2. Extension of Prior Work: The authors improve upon simpler one-layer linear models from prior work by introducing a more sophisticated two-layer model. This extension potentially can provides deeper insights into the scaling behavior of neural networks.

**Weaknesses:**

1. Complexity and Assumptions: The paper is difficult to follow due to the heavy reliance on prior work. This reliance can make it challenging for readers who are not deeply familiar with the specific foundational literature to fully grasp the contributions.

2. Unclear Definition of Feature Learning: Throughout the paper, the concept of "feature learning" is not clearly defined. It remains unclear how the authors measure feature learning or provide a tangible understanding of it. Although allowing
A(t) to evolve leads to better performance, this change doesn’t adequately explain how features are being learned or evolved.

3. Limited Model Scope: The model seems quite limited in its design. Allowing
A(t) to change is reasonable, especially since the target function is 𝑊∗𝜓_{inf}(x). However, it is unclear why this is believed to represent or mimic feature learning in practical, real-world scenarios. The connection to real feature learning remains vague. This is also related to 2nd point, that I do not understand how authors define feature learning.

4. Transferability to Deep Nonlinear Models: It is not clear why the results derived from the proposed linear model are expected to transfer well to deep, nonlinear models.

overall I feel like the paper is scattered and I can not grasp the main point authors are trying to make about feature learning.

**Questions:**

See weaknesses

---

> ### Author Response · Authors · 2024-11-20
> **Response**
>
> We have tried addressing the concerns and questions raised by this reviewer and hope that in light of these they would consider increasing their score.
>
> ### Strengths:
> *Tractable Theoretical Setting: The paper presents a solvable theoretical model, which allows for potentially clearer and rigorous exploration of neural scaling laws beyond the kernel regime. This tractable setting potentially makes it easier to understand and predict the dynamics of feature learning in neural networks.*
>
> *Extension of Prior Work: The authors improve upon simpler one-layer linear models from prior work by introducing a more sophisticated two-layer model. This extension potentially can provides deeper insights into the scaling behavior of neural networks.*
>
> Thank you for appreciating these aspects of our work.
>
> ### Weaknesses
>
> *Complexity and Assumptions: The paper is difficult to follow due to the heavy reliance on prior work. This reliance can make it challenging for readers who are not deeply familiar with the specific foundational literature to fully grasp the contributions.*
>
> We have tried to improve some of the exposition and motivate the model (see blue text) without relying as much on prior works. We also removed some unnecessary mathematical detail from the main text.  We can make additional improvements before the final version of the paper.
>
> *Unclear Definition of Feature Learning: Throughout the paper, the concept of "feature learning" is not clearly defined. It remains unclear how the authors measure feature learning or provide a tangible understanding of it. Although allowing A(t) to evolve leads to better performance, this change doesn’t adequately explain how features are being learned or evolved.*
>
> We will give a clear operational definition of feature learning: the empirical NTK of the model changes. We state this in the first paragraph of section 2.
>
> *Limited Model Scope: The model seems quite limited in its design. Allowing A(t) to change is reasonable, especially since the target function is 𝑊∗𝜓_{inf}(x). However, it is unclear why this is believed to represent or mimic feature learning in practical, real-world scenarios. The connection to real feature learning remains vague. This is also related to 2nd point, that I do not understand how authors define feature learning.*
>
> This is a fair critique. However, we note that this modeling assumption is basically saying that the empirical NTK can evolve throughout training. The $A(t)$ matrix gives the coefficients for the empirical kernel features at time $t$ in terms of the infinite kernel features at time $t=0$. This matrix exists for deep nonlinear networks, but we do not know apriori how it evolves. This motivated us to consider a simple update rule for $A$.
>
> *Transferability to Deep Nonlinear Models: It is not clear why the results derived from the proposed linear model are expected to transfer well to deep, nonlinear models.*
>
> This is a good question. While our theory does not directly characterize nonlinear networks, we wanted to perform some empirical tests in more realistic settings where we could examine how general our model's predictions were. We suspect that the finite/infinite RKHS norm as a transition point for feature learning since perturbations around the kernel limit are **unstable** if the target function is outside of the RKHS of the initial kernel (if $\beta<1$). The fact that our new predicted exponent $2\beta/(1+\beta)$ still seems to work on nonlinear tasks is fortunate, but not guaranteed by our theory and is currently conjectural.
>
>
> *overall I feel like the paper is scattered and I can not grasp the main point authors are trying to make about feature learning.*
>
> Thank you for pointing this out. We will try stressing the main points about feature learning which are
> 1. Feature learning may only change scaling laws for tasks which are sufficiently hard for the initial kernel.
> 2. In our model with natural power law covariate structures with the target function outside the RKHS, feature learning improves the scaling exponents for optimization but not with respect to model size.
> 3. The scaling law for SGD in this model with respect to training time improves from $t^{-\beta}$ to $t^{-2\beta/(1+\beta)}$ while the parameter scaling laws do not change.
>
> We added the following sentence below our listed contributions
>
> "Overall our results suggest that feature learning may improve scaling law exponents by changing the optimization trajectory for tasks that are hard for the initial kernel."

---

> > ### Author Response · Authors · 2024-11-25
> > **Following up**
> >
> > We are just following up on this to see if our revisions and comments answered the questions and concerns posed by this reviewer. Please let us know if anything remaining needs to be clarified.

---

> > > ### Comment · Reviewer_tuhi · 2024-11-26
> > >
> > > Thank you for the clarifications.
> > >
> > >
> > > I still have the following concerns,
> > >
> > > >We will give a clear operational definition of feature learning: the empirical NTK of the model changes. We state this in the first paragraph of section 2.
> > >
> > > 1. This definition does not seem appropriate. I believe what you intended to convey is the alignment of the empirical NTK with the ground truth NTK, which I assume is known due to the setup assumptions. If this is the case, we need to investigate how this alignment evolves during training. Does it improve consistently, or does it oscillate?
> > >
> > > 2. Given this new definition of feature learning, I am now unclear on how this can be generalized to tasks like CIFAR and MNIST. Could you explain how the evolving empirical NTK relates to image classification tasks? Specifically, which empirical NTK are we trying to align with in these cases?

---

> > > > ### Author Response · Authors · 2024-11-26
> > > > **Response to Follow up**
> > > >
> > > > We thank the reviewer for the follow up questions and for allowing us to clarify our setup.
> > > > 1.  We take feature learning to mean that the neural network operating outside of the lazy training regime, agnostic to how the NTK evolves. This is consistent with the definitions used in many prior works in this area (https://arxiv.org/abs/1906.08034, https://arxiv.org/abs/2106.10165 , https://arxiv.org/abs/2011.14522, https://arxiv.org/abs/2309.16620 , to share just a few ).  However, we do note that our theory **does predict** task-relevant adaptation of the NTK, which aligns to the target function. The NTK of our student model has dynamics that are given in equation 52. If $\gamma = 0$, then the hidden weights A and the kernel K are both frozen. For nonzero $\gamma$ these can both evolve. While we do not have a "ground truth NTK" in our setting, we do have a target function y(x) that can be completely generic. The NTK of the model evolves and partially aligns to this fixed target function over training.
> > > > 2. Since our theory can handle generic target functions $y(x)$, we can apply it on real datasets. Practically, we take the real labels y from the data and decompose them in the eigenbasis of the initial infinite-width kernel. This gives us the coefficients $w_\star$ that are inputs to the theory.  We show these spectral diagonalizations in Figure 8.  We show the rate of decay of the eigenvalues and the task coefficients $w_\star$ in Figure 8 b. At $t=0$, the rate of decay implies that $\beta \approx 0.15$. We also plot the decomposition of the target labels in the eigenbasis of the **evolving empirical kernel** and we see that a large fraction of task variance is captured by the top eigenmode, quantifying the alignment of the kernel with the task.

---

> > > > > ### Comment · Reviewer_tuhi · 2024-11-26
> > > > >
> > > > > Thank you for the clarification.
> > > > >
> > > > > I still have significant concerns about the "feature learning" part. This paper is about how "feature learning" can potentially help with the scaling law, yet there is no definition of "feature learning" in the original text. Now, authors have provided two definitions:
> > > > >
> > > > > >1. "We will give a clear operational definition of feature learning: the empirical NTK of the model changes. We state this in the first paragraph of section 2."
> > > > > >2. "We take feature learning to mean that the neural network operates outside of the lazy training regime, agnostic to how the NTK evolves."
> > > > >
> > > > > First, these two definitions are different. How can a paper be about "feature learning" but lack a clear, consistent definition? This aside, both attempts fail to provide a meaningful definition.
> > > > >
> > > > > Here’s why I did not find these definitions useful:
> > > > >
> > > > > 1. If I randomly change the parameters of my model, the empirical NTK would change drastically. However, no useful features would emerge from such a change. Therefore, stating that "the empirical NTK of the model changes" as a definition of feature learning is neither meaningful nor sufficient.
> > > > > 2. How can the definition of feature learning depend on the empirical NTK while simultaneously claiming to be agnostic to its evolution? What’s missing is a clear explanation of why the NTK’s evolution represents a good evolution—one that leads to meaningful feature learning. While improvements in loss might hint at beneficial changes, they do not serve as concrete evidence of feature learning.
> > > > >
> > > > >
> > > > > Overall, I am still unclear about how you define and measure feature learning. It seems there is no clear metric or approach for demonstrating it so far. This issue is particularly critical because it affects the clarity of your main contributions. Here’s what you state as your key findings to answer my first comment:
> > > > >
> > > > > > We will try stressing the main points about feature learning which are:
> > > > > >1. Feature learning may only change scaling laws for tasks which are sufficiently hard for the initial kernel.
> > > > > >2. In our model with natural power law covariate structures with the target function outside the RKHS, feature learning improves
> > > > > >3. the scaling exponents for optimization but not with respect to model size.
> > > > >
> > > > > In point 1, it is claimed that feature learning affects scaling laws, but what does that mean? Based on the provided definition, if I randomly change the weights of my model, I can significantly alter the empirical NTK, but this will not affect scaling laws in any meaningful way. In point 2, it is stated that "feature learning improves" scaling exponents—but what does that improvement actually mean? Why not simply say the test loss decreases or the performance improves?
> > > > >
> > > > > Without a clear definition or measure of feature learning, it is not possible to understand the claims.

---

> > > > > > ### Author Response · Authors · 2024-11-27
> > > > > > **Response to Reviewer Questions**
> > > > > >
> > > > > > We would like to clarify any confusion about our claims and possible definitions of feature learning.
> > > > > >
> > > > > > The two definitions we provided previously are indeed compatible with each other as follows: in lazy training, the empirical NTK (eNTK) of a given network does not change. In the feature learning regime, as defined by the references cited above, is precisely when the eNTK actually changes. This is the first definition that you cite, and also the first clause of the second definition. The statement “agnostic to how the NTK evolves” simply states that we are not putting any stringent conditions on the nature of this evolution to define feature learning. The two definitions are indeed compatible. We note that this definition depends on the NTK while being agnostic to the specific details of its evolution. There is no contradiction.
> > > > > >
> > > > > > However, as the reviewer points out, one might want a more refined definition of feature learning beyond just “the network is not in the lazy/kernel regime”. The random parameter change is a good example. Indeed, one can quantify how “useful” the representation of a given network is by calculating how the kernel aligns to task relevant information (cosine similarity between the kernel $K$ and the labels, or the cumulative power decomposition in Figure 8b). Having improved kernel alignment can be viewed as the network learning “useful features”. In our toy model, the eNTK changes and the alignment improves, so the network learns useful features as well. Improved kernel alignment has also been observed in realistic networks ( [here](https://arxiv.org/abs/2008.00938), [here](https://arxiv.org/abs/2010.15110),  [here](https://ieeexplore.ieee.org/abstract/document/9929375), to share just a few). We will add this expanded discussion to the main text if it would assuage concerns.
> > > > > >
> > > > > > This paper gives a concrete example of how when a network’s empirical neural tangent kernel is allowed to evolve with a reasonable SGD update, it can improve the scaling law exponent of generalization performance with time. This effect has not been studied in prior works. It is indeed the case that not only does the kernel evolve, but the kernel alignment improves as well. We do not claim that our toy model is fully representative of realistic networks, and indeed we have expounded on its limitations. Nonetheless, one can indeed observe an improvement consistent with our scaling law exponents for some nonlinear networks trained on realistic data, such as an MLP on MNIST-1M or the early stage training of a CNN on CIFAR-5M.
> > > > > >
> > > > > > We do not claim that feature learning always improves scaling laws. The counterexample that you point out is one such example. Indeed, a large portion of our paper shows that in our model, the movement of the kernel is not sufficient to improve the scaling law exponent. The key result of our work is that when the task is “hard” as measured by the source exponent, then a linear network will encounter an improved scaling law. Because one can define source exponents for nonlinear models as well by measuring the decay exponent of the task in the eNTK eigenbasis, one can extend this to a hypothesis (which we term the source hypothesis) for nonlinear networks.
> > > > > >
> > > > > > We hope that this is helpful and assuages some of the reviewer’s concerns. We have added an appendix section H on the definition of feature learning in an attempt to engage with this discussion in good faith.

---

> ### Comment · Reviewer_tuhi · 2024-12-03
>
> Thank you for providing additional clarifications and for the efforts to expand the discussion in Appendix H. I appreciate the engagement and the inclusion of definition B and task alignment, which indeed offer a meaningful direction to explore the concept of feature learning further.
>
> >The two definitions we provided previously are indeed compatible with each other..."
>
> I did not mean to suggest that the definitions were contradictory. However, I maintain that they are distinct in focus and scope. The definition of feature learning as "the empirical NTK of the model changes" seems overly broad and potentially misleading. While the second definition, which incorporates the lazy regime and optimization perspectives, offers a more refined approach, it still falls short in directly addressing the quality or utility of the features learned. In particular, observing changes in the loss function alone does not provide sufficient evidence to conclude that meaningful or task-relevant features are being learned.
>
> That said, the added Appendix H is a promising and commendable addition. Specifically, definition B and the concept of task alignment resonate well with the broader understanding of feature learning. I would be particularly interested in seeing how alignment evolves across tasks of varying hardness, especially for tasks with β<1. Unfortunately, the main plots in the paper predominantly focus on the loss function, which does not provide much insight into the dynamics or utility of feature learning.
>
> I recognize the contribution of the paper in demonstrating the interplay between task hardness and the lazy or rich regime scaling law, and Table 1 serves as an excellent summary in this regard. However, the framing around feature learning remains problematic. The results and analyses in the paper could be entirely framed around the lazy and rich regimes without invoking the concept of feature learning.
>
> In light of this, I would strongly recommend either (1) revising the framing of the paper to remove claims about feature learning, or (2) building upon definition B to empirically study alignment changes across different tasks. Without either of these adjustments, the connection between the presented results and feature learning remains tenuous and detracts from the otherwise valuable contributions of the paper.
>
> >We do not claim that feature learning always improves scaling laws...
>
> This is an important point, but I think what this paper shows is that being in the rich regime does not always help scaling laws, and indeed the lazy regime might be just as good. This is probably related to the feature learning and alignment argument, but with the current results, I am not sure what we can say about feature learning in the sense of definition B in Appendix H.
>
> **Final word:**  I understand that it is too late to make significant changes, but it would be great to see the alignment evolution explored further using definition B. That said, I strongly suggest adding definition A earlier in the paper for clarity. Based on definition A, feature learning is simply defined as being in the rich regime and allowing the weights to change. While I do not believe this definition is reasonable, I understand that it has unfortunately been repeatedly used in prior literature. In light of this, I have decided to increase my score to 6.

---

### Official Review · Reviewer_6VnB · 2024-11-05

**Soundness:** 4
**Presentation:** 3
**Contribution:** 4
**Rating:** 8
**Confidence:** 4

**Summary:**

The paper does a mean-field analysis of feature learning in a model similar to a shallow linear network. Scaling laws are obtained, where depending on the source exponent, feature learning can lead to a improvement in the exponent.

**Strengths:**

The paper uses advanced theoretical tools to tackle the important question of the effect of feature learning. A complete phase diagram is obtained, showing under which condition feature learning leads to a significant advantage. The theoretical scaling laws are checked empirically, and seem to even extend to some degree to deep nonlinear networks (which are not covered by the theory).

**Weaknesses:**

The paper does not give much intuition for why this improvement of feature learning is observed, instead the authors simply mention the existence of a self-consistent dynamic derived in the appendix.

I also feel that the authors do not relate much to the already existing litterature on the dynamics of linear networks (e.g. https://arxiv.org/abs/2211.16980), where feature learning and self-consistent dynamics have been described in https://arxiv.org/abs/2406.06158 or https://arxiv.org/abs/2405.17580 . It would be interesting to know whether the dynamics observed in this paper is similar to the ones observed in these papers.

**Questions:**

The effect observed in this paper differs significantly from other feature learning paper in that it does not assume any form of sparse structure (actually feature learning appears to help when the signal is closest to being of the same intensity along all directions), i.e. low-index models for shallow networks, or low-rank matrices for linear networks. This suggest that the type of feature learning observed here is quite different from these other line of results. My intuition from your paper, is that feature learning may in some sense improve the conditioning of the Hessian/NTK, allowing faster convergence, whereas in the presence of sparsity feature learning seems to rather make the NTK low-rank and more ill-conditioned. I was therefore wondering whether the effects observed here might disappear when switching from online learning to GD on a fixed dataset, because it would untangle the training speed from the number of datapoints observed, and maybe we could see there that feature learning makes convergence faster but maybe one still converges to approximately the same solution (and thus same test error).

Typos:
- on page 5: The sentence "The first terms represent bottleneck/resolution-limited scalings in the sense that taking all other quantities to infinity and studying scaling with the last ..." does not finish.

---

> ### Author Response · Authors · 2024-11-20
> **Response**
>
> Thank you for the careful reading and comments. We have tried addressing these concerns below.
>
> ### Strengths:
> *The paper uses advanced theoretical tools to tackle the important question of the effect of feature learning. A complete phase diagram is obtained, showing under which condition feature learning leads to a significant advantage. The theoretical scaling laws are checked empirically, and seem to even extend to some degree to deep nonlinear networks (which are not covered by the theory).*
>
> We thank the reviewer for their careful reading and for appreciating these strengths of our work.
>
> ### Weaknesses
>
> *The paper does not give much intuition for why this improvement of feature learning is observed, instead the authors simply mention the existence of a self-consistent dynamic derived in the appendix.*
>
> We thank the reviewer for this point. We will revise the draft to provide more intuition for the effect. The main phenomenon is that, for tasks outside of the RKHS of the initial kernel, the empirical kernel will grow as a power law, driving down the loss with an improved rate.
>
> We added some text in the section titled "Accelerating Training in Rich Regime" which states
>
> "This acceleration is caused by the fact that the effective dynamical kernel $K(t)$ diverges as a powerlaw $K(t) \sim t^{1-\chi}$ when $\beta < 1$ (see Appendix \ref{app:bottleneck_scalings}). This is due to the fact that the kernel approximation at finite $\gamma$ is not stable when training on tasks out of the RKHS."
>
>
> *I also feel that the authors do not relate much to the already existing literature on the dynamics of linear networks (e.g. https://arxiv.org/abs/2211.16980), where feature learning and self-consistent dynamics have been described in https://arxiv.org/abs/2406.06158 or https://arxiv.org/abs/2405.17580 . It would be interesting to know whether the dynamics observed in this paper is similar to the ones observed in these papers.*
>
> We appreciate the reviewer for pointing us to these relevant works. The feature learning papers that examine lazy/active regimes are related to our $\gamma$ parameter and we will mention this relationship (specifically $\gamma \to 0$ gives a lazy limit like $1/\tau$ in the Kunin et al paper). Specifically, the hidden weights $A$ in our model are updated at a relative rate $\gamma$ compared to the readout weights, which is an *unbalanced* case considered by Kunin et al. Our dynamics can also be viewed as "mixed" in the sense of Tu et al and their Markovian dynamics closely resemble the dynamics we obtain in the $N \to \infty$ limit (Appendix E.1.2). Lastly, just like the Chizat paper, we also would observe the $1/\text{width}$ convergence rate to the limiting large width dynamics at very early times or in the overparameterized regime when data are repeated. However, our model also captures finite width effects and can exhibit *power law* model-bottleneck scaling laws at late time which are task-dependent.
>
> Some things that differ between our analysis and some of these prior works are
> 1. Allows for arbitrary (and as special case, power law) feature structure. We solve for the resulting power law exponents for the loss.
> 2. Our projected gradient descent can exhibit power law scaling with the width $N$ of the network (not just $1/N$ as is reported in Tu et al), recovering a Chinchilla-like compute optimal scaling law.
> 3. We also consider the impact of SGD noise and show it only alters the scaling law picture in the super easy task regime, even with feature learning.
>
> We added a paragraph in the Related works section that explicitly references these three works.
>
> "Recent works have examined the dynamics of linear networks, contrasting the dynamics in lazy and feature learning regime, including analysis of infinite width linear networks \cite{chizat2024infinite}, and linear networks with varying and unbalanced initialization and learning rates \cite{kunin2024get, tu2024mixed}. Our model can be interpreted as a two-layer linear network which captures finite width effects (with task-dependent scaling laws) from random initialization. Like these related works, our model also has unbalanced learning rates between hidden and readout weights set by a parameter $\gamma$ that recovers a lazy limit as $\gamma \to 0$."

---

> > ### Author Response · Authors · 2024-11-20
> > **Response Part 2**
> >
> > ### Questions:
> > *The effect observed in this paper differs significantly from other feature learning paper in that it does not assume any form of sparse structure (actually feature learning appears to help when the signal is closest to being of the same intensity along all directions), i.e. low-index models for shallow networks, or low-rank matrices for linear networks. This suggest that the type of feature learning observed here is quite different from these other line of results. My intuition from your paper, is that feature learning may in some sense improve the conditioning of the Hessian/NTK, allowing faster convergence, whereas in the presence of sparsity feature learning seems to rather make the NTK low-rank and more ill-conditioned. I was therefore wondering whether the effects observed here might disappear when switching from online learning to GD on a fixed dataset, because it would untangle the training speed from the number of datapoints observed, and maybe we could see there that feature learning makes convergence faster but maybe one still converges to approximately the same solution (and thus same test error).*
> >
> >
> > This is a great question. We do think that the nature of feature learning in this setting is different than the low-rank tasks that are often considered in the literature. Also the reviewer's intuition about sample complexity is correct. We can consider data repitition (Equation 41) and argue in Appendix E.4 that the data scaling laws are *unchanged* compared to the lazy learning limit. We think it is an important open question whether *sample scalings* can ever be improved on these tasks with power law structure which are closer to those observed in natural data.

---

> > > ### Comment · Reviewer_6VnB · 2024-11-25
> > >
> > > I thank the authors for the detailed answer and for adding comparisons to the few papers I mentioned. I am keeping my score, this is a good paper.

---

### Author Response · Authors · 2024-11-20
**Global Response**

We thank the reviewers for their detailed comments. Based on some recurring concerns and questions, we have made the following adjustments to the paper (blue text in the new draft).

1. We now provide more comparison to relevant linear network analyses in lazy and rich regimes. This new paragraph is at the end of the related works section.
2. We have tried clarifying our operational definition of feature learning. In this work, we consider feature learning to be evolution of the NTK (which is certainly a necessary condition of feature learning). We are aware that other definitions may be possible, but this is what we are focused on in the present work.
3. We provide more explanations and predictions of the impact of $\gamma$ on scaling law, specifically the time it takes to transition to the new scaling behavior and the multiplicative reduction in loss that depends on $\gamma$.
4. We provide more information about the data repetition setting and in the Appendix compare the data scaling law with the lazy case, finding that the exponents are the same in lazy and rich regimes much like the $N$ scaling.
5. We now provide a more comprehensive comparison to prior works in a new Appendix section, specifically Bordelon et al 2024 and Paquette et al 2024.
6. Overall, we have generally aimed to provide more helpful detail and motivation for our model throughout the main text.

We hope that in light of these updates and our responses to individual reviewer comments that the reviewers will consider favoring acceptance. We are happy to make additional adjustments to the paper or to answer any additional questions during the remaining discussion period.

---

### Meta-Review · Area_Chair_DFiW · 2024-12-16

**Metareview:**

This paper studies the scaling laws of a solvable two-layer linear model trained by a variant of SGD in two regimes: lazy learning and feature learning (Strictly speaking, three scaling schemes, hard, easy, super easy based on the source condition). Based on the derivation, the paper gives the power-law variation of the loss with training time ( t ), number of parameters ( N ), and batch size ( B ) in these two regimes. The main contribution of this work shows that feature learning can accelerate the convergence of hard tasks.

The AC also reads this paper. The analysis starts from NTK and then considers the trainable features in Eq. (3) by SGD variants. This scheme indeed simpifies the analysis but is a bit far away from standard neural network training. The key analysis for scaling law used in this paper is based on DMFT, and then the scaling law can be given under souce and capacity conditions. In the AC's view, this paper uses a "tricky" way from the kernel regime to deliver the result for neural networks. But this is still a good paper to the community, so I recommend to accept this paper as a poster.

Furthermore, the AC suggests the author to carefully polish the paper and re-organize the structure for readability, as well as the discussion about related work and a mathematical definition of feature learning.

**Additional Comments On Reviewer Discussion:**

All of the reviewers engaged with the discussion, and most of issues are clarified.
There are some issues, e.g., mathematical definitions of feature learning raised by the reviewers.
We think that the authors can fix it well in the updated version.

---

### Decision · Program_Chairs · 2025-01-22

Accept (Spotlight)